# Tetrahydroquinoline-Isoxazole/Isoxazoline Hybrid Compounds as Potential Cholinesterases Inhibitors: Synthesis, Enzyme Inhibition Assays, and Molecular Modeling Studies

**DOI:** 10.3390/ijms21010005

**Published:** 2019-12-18

**Authors:** Yeray A. Rodríguez Núñez, Margarita Gutíerrez, Jans Alzate-Morales, Francisco Adasme-Carreño, Fausto M. Güiza, Cristian C. Bernal, Arnold R. Romero Bohórquez

**Affiliations:** 1Laboratorio Síntesis Orgánica, Doctorado en Ciencias Mención Investigación y Desarrollo de Productos Bioactivos, Instituto de Química de Recursos Naturales, Universidad de Talca, Casilla 747, Talca 3460000, Chile; yenuro30@gmail.com; 2Laboratorio Síntesis Orgánica, Instituto de Química de Recursos Naturales, Universidad de Talca, Casilla 747, Talca 3460000, Chile; 3Center for Bioinformatics, Simulations and Modeling, Faculty of Engineering, University of Talca, 2 Norte 685, Casilla 721, Talca 3460000, Chile; jalzate@utalca.cl (J.A.-M.); francisco.adasme@gmail.com (F.A.-C.); 4Grupo de Investigación de Compuestos Orgánicos de Interés Medicinal (CODEIM), Parque Tecnológico Guatiguará, Universidad Industrial de Santander, A.A. 678, Piedecuesta 681011, Colombia; fausto.marin1@correo.uis.edu.co (F.M.G.); cristian.bernal@correo.uis.edu.co (C.C.B.)

**Keywords:** Alzheimer’s disease, cholinesterase inhibitors, cross-docking and MM/GBSA free binding energy, hybrid compounds

## Abstract

A series of 44 hybrid compounds that included in their structure tetrahydroquinoline (THQ) and isoxazole/isoxazoline moieties were synthesized through the 1,3-dipolar cycloaddition reaction (1,3-DC) from the corresponding *N*-allyl/propargyl THQs, previously obtained via cationic Povarov reaction. In vitro cholinergic enzymes inhibition potential of all compounds was tested. Enzyme inhibition assays showed that some hybrids exhibited significant potency to inhibit acetylcholinesterase (AChE) and butyrylcholinesterase (BChE). Especially, the hybrid compound 5n presented the more effective inhibition against AChE (4.24 µM) with an acceptable selectivity index versus BChE (SI: 5.19), while compound 6aa exhibited the greatest inhibition activity on BChE (3.97 µM) and a significant selectivity index against AChE (SI: 0.04). Kinetic studies were carried out for compounds with greater inhibitory activity of cholinesterases. Structure–activity relationships of the molecular hybrids were analyzed, through computational models using a molecular cross-docking algorithm and Molecular Mechanics/Generalized Born Surface Area (MM/GBSA) binding free energy approach, which indicated a good correlation between the experimental inhibition values and the predicted free binding energy.

## 1. Introduction

Alzheimer’s disease (AD) is the most common form of dementia that affects over 30 million people, most of whom are elderly [1]. AD is a major public health problem in developed countries, which is characterized by loss of memory and the gradual decline in cognitive functions resulting in behavior changes [2]. Knowledge about its causes and mechanisms has grown enormously in the last decades. For instance, extracellular deposits of β-amyloid plaques (Aβ), the formation of intracellular neurofibrillary tangles (NFT), decreased activity of choline acetyltransferase enzyme, and oxidative stress are some of the pathological features of AD [3,4].

There is currently no cure for AD, although there are some drugs available that can delay the disease progression at certain stages of illness [5]. The most significant strategy of palliative treatment for AD has been the use of cholinergic drugs, which counteract cognitive dysfunction by inhibition of the enzyme AChE, thus increasing the levels of the neurotransmitter acetylcholine, regulating synaptic functions [6].

The active site of AChE (CAS) is composed of a catalytic triad (Ser-His-Glu), found at the bottom of ~20 Å deep gorge lined by a high presence of aromatic residues [7], and another binding site named the peripheral anionic site (PAS) that is located at 18 Å away from the active site [8]. BChE is an enzyme with a supportive role in the cholinergic process. It has received important attention because once it is inhibited, cognitive functions are improved [9]. Although AChE and BChE share some structural similarities, including the presence of a catalytic center within the gorge, ligand binding modes of BChE differ significantly from those of AChE due to the absence of some aromatic residues located mainly in the acyl binding pocket at midgorge. Thereby, different compounds can specifically inhibit one of those enzymes, or they are also able to exert a dual binding action, depending on amino acid recognition and interactions generated within the active site [10,11].

Commercial drugs like donepezil, rivastigmine, galantamine, and tacrine have demonstrated efficacy in the treatment of patients with a mild-to-moderate AD, although the therapeutic use of tacrine has been discouraged due to its hepatotoxicity effects [12]. Generally, these drugs bind at a specific point of the active site, but recent research is looking for compounds having an action at several points of the active site in order to increase their bioactivity. Seeking to obtain new compounds that can recognize more than one binding spot in the same therapeutic target, tacrine, galantamine, and donepezil have been used as scaffolds to produce a new series of hybrid compounds for improving the inhibitory activity against cholinergic enzymes, and at the same time, improve their pharmacological properties [13,14,15,16]. Curcumin–tacrine hybrids have been synthesized as multifunctional compounds that interact with the catalytic active site and the peripheral anionic site of AChE, showing positive inhibitory effects even more potent than individual tacrine [17]. Similarly, the synthesis of molecular hybrids from donepezil and ebselen, a drug capable of inhibiting iron-induced tau phosphorylation, has been described. The new hybrid compounds showed good dual inhibitory activity against AChE and BChE [16]. Thus, the molecular hybridization synthetic approach can be applied to obtain new molecules from the combination of two or more pharmacophoric structures that are present in bioactive compounds [18] (Figure 1).

With all the above in mind, our previously reported work has shown the synthesis and anticholinesterase activity of *N*-allyl/propargyl THQs obtained by an efficient and mild methodology based on the cationic Povarov reaction (a domino Mannich/Friedel–Crafts reaction) [19] and cytotoxicity on HeLa and HepG2 cancer cell for some THQ–isoxazole hybrids [20,21]. As a further step in the development of new molecules with improved cholinergic activity, in the present study, we describe the successful synthesis of THQ–isoxazoline/isoxazole hybrids through the 1,3-DC reaction, as well as their in vitro inhibitory activity evaluation against AChE and BChE. Additionally, their ADME-Tox properties (absorption, distribution, metabolism, excretion, and toxicity), potential binding modes, and binding affinity against AChE and BChE were predicted using computational tools like ensemble docking and MM/GBSA calculations.

## 2. Results and Discussion

### 2.1. Chemistry

Taking advantage of the reactivity of the propargyl and allyl fragments present in the THQ compounds that were previously synthesized via the cationic Povarov reaction [19], the synthesis of new THQ–isoxazoline/isoxazole hybrids was carried out using a gentle and efficient process based on 1,3-dipolar cycloaddition. In this reaction, nitrile oxides (formed in situ from corresponding aldoximes) were used as dipole and the propargyl and allyl THQs as dipolarophiles. In both cases, reactions occur easily and with good yields in anhydrous DCM in the presence of Et_3_N and NaOCl at room temperature (Scheme 1). New hybrid compounds THQ–isoxazoline (series 5) and THQ–isoxazole synthesized (series 6) contain a methylene bridge that keeps the two N-heterocyclic rings together.

The hybrid compounds of series 5a-p were obtained as a mixture of a pair of non-separable diastereoisomers through column chromatography. In this case, all compounds are viscous oils, and the reaction takes place with good yield after purification (63–77%; Appendix A). Whereas, hybrid compounds of series 6a-ab were obtained mostly as viscous oils, and some as stable solids, with good and excellent reaction yields after column chromatography (40–95%; Appendix A).

In this report, 40 hybrid compounds with four (4) different substituent groups at the C-6 position of the THQ scaffold were synthesized. In addition, when the aryloxime 2 was 3,4-dimethoxybenzaldehyde oxime, four other (4) new hybrid compounds with substituent groups at C-6 and C-8 positions of THQ were obtained. All THQ–isoxazole/isoxazoline molecular hybrids were obtained as regioisomers 3,5-disubstituted and in all of them were possible systematically characterized by spectroscopic techniques, including IR, ESI-MS, ^1^H-NMR, and ^13^C-NMR.

In the case of IR spectra, typical bands for carbonyl group C=O were observed (1656–1681 cm^−1^). The band associated with the vibration of the C–O bond of methoxy groups from the starting aldoximes (1122–1178 cm^−1^) was also observed. Besides, the vibration band of the aromatic C–H bonds at 2832–2976 cm^−1^ was identified easily. ^1^H NMR and ^13^C NMR spectroscopy in CDCl_3_ solution were carried out, and all signals for each individual H–atom and C–atom were properly assigned. All ^1^H-NMR spectra of the THQ–isoxazole/isoxazoline hybrids synthesized showed a similar patron, which was characterized by the presence of several well-defined signal groups. At high field (chemical shifts between 1.80–3.60 ppm) the signals corresponding to the methylenic protons of the 2-pyrrolidone fragment and the piperidine ring were characterized. The signal corresponding to the chiral proton of the THQ ring was observed around 5.40 ppm. Signals for aromatic protons were observed and clearly differentiated between 6.31–7.70 ppm. In the case of molecular hybrids belonging to the THQ–isoxazole 6 series, a single signal was observed between 6.29–6.63 ppm, corresponding to the proton of the isoxazole ring, which gives strong evidence of the successful formation of the new heterocyclic. Similarly, THQ-isoxazoline 5 molecular hybrids showed a signal around 4.89–5.21 ppm indicating the formation of the isoxazoline ring. Furthermore, the lack of signal for protons from the propargyl and the allyl fragments was additional evidence that the 1,3-DC reaction was carried out satisfactorily.

### 2.2. Cholinesterase Biological Activity 

All synthesized hybrid compounds were evaluated in vitro as dual AChE/BChE inhibitors. The IC_50_ values were calculated by means of regression analysis. Results are expressed as averaged μM ± SEM values, and compared using ANOVA analysis. A *p*-value of less than 0.05 was considered significant. Details for pharmacological experiments are described in the experimental section as well as in our previous report [19]. 

Results of enzyme inhibition assays of THQ–isoxazoline hybrids 5a-p on AChE and BChE showed that the hybrid 5k (R_1_ = OCH_3_) and hybrids 5n, 5o, and 5p (R_1_ = Cl) presented good inhibitory action against AChE (IC_50_ value < 15.26 μM). Compound 5n is highlighted with an IC_50_ value of 4.24 μM. These THQ-isoxazoline hybrids also showed high selectivity for AChE enzyme over BChE (SI values of 16.8, 5.19, 6.35, and 8.61, respectively) (Table 1). 

Conversely, results indicated that THQ–isozaxole hybrids 6a (R_1_ = H), 6h (R_1_ = OCH_3_), 6r (R_1_ = CH_2_CH_3_), and hybrids 6z and 6aa (R_1_ = Br) showed a very high inhibitory activity against BChE, with IC_50_ values (<7.18 μM) even better than that found for the reference compound galantamine (IC_50_ = 8.80 μM), a commercial drug widely used in the palliative treatment of AD. Hybrid 6aa, a molecular hybrid with a bromine atom at C-6 in the THQ ring and 3,4-dimethoxyphenyl group at the C-3 of isoxazole moiety, was the most active compound (IC_50_ = 3.97 μM). Notably, this hybrid compound showed high selectivity for BChE, being about 24 times more potent against BChE than for the AChE enzyme (Table 2).

As can be seen (Table 1 and Table 2), a significant decrease in IC_50_ values and consequently a better inhibitory activity of these hybrid compounds are observed when different substituents groups are introduced into the THQ scaffold, especially halogens (Cl and Br), and methoxy groups into aryl fragment on C-3 position of isoxazoline or isoxazole ring. However, beyond the different substituents groups on the hybrid molecules and their impact on IC_50_ values, it is important to rescue the synergistic effect generated by the two fused heterocyclic rings, in order to inhibit the biological activity of cholinergic enzymes selectively. Most of the new hybrid compounds turned out to be effective inhibitors of cholinergic enzymes with IC_50_ values spanning a μM range. The new designed hybrid compounds are better inhibitors of cholinergic enzymes when compared to our previous series [19] and, in some cases, more active than the reference drug, galantamine.

Based on the previously obtained results, and with the aim to assess the kinetic model of AChE/BChE inhibition of the target compounds, the most active compounds 5n and 6aa were subjected to kinetics studies. For this purpose, the rate of enzyme activity was measured at eight different concentrations of substrate ATC/BTC. The obtained Michaelis–Menten parameters along with the Lineweaver–Burk plots allow us to conclude that the compound 5n (Figure 2a) displayed a mixed-type inhibition against AChE with variation in Km and Vmax values. However, the compound 6aa (Figure 2b) showed a competitive type inhibition against BChE with variation in Km values.

The analysis of inhibition modality is a piece of relevant information in the early stages of design and drug discovery, showing an approximation to the type of bind with the active site. As an example, competitive inhibitors bind exclusively to the free enzyme form, while non-competitive or mixed type inhibitors bind with some affinity to the free enzyme, as the enzyme–substrate complex. Thus, the non-competitive modality can be an advantage in vivo when the physiological context exposes the enzyme to high substrate concentrations. Although the clinical advantage of non-competitive inhibition has been recognized, there are a very large number of drugs in clinical use today that are competitive enzyme inhibitors, and which can be related to historical approaches for drug discovery that have been focused on active site-directed inhibitors [22,23]. Here, we presented two hybrids from tetrahydroquinolines that act via competitive (6aa) and mixed (5n) mechanism. The mechanistic differences between compounds 5n, 6aa, and galantamine could be assigned to the π–π interactions with Trp84, Tyr334, and Phe330, which were observed only for these two tetrahydroquinolines derivatives with higher inhibitory potency. These results highlight the importance of the π–π interaction for the ligands and thus can be used for designing new AChEIs.

### 2.3. Binding Modes and Affinities Against AChE

The binding modes of the synthesized compounds within the AChE active site were predicted by means of ensemble docking, where each ligand is docked into multiple protein structures, as a low-cost alternative to model protein flexibility during docking. As the chirality of the scaffold was unknown, all possible enantiomers (120 in total) were considered in the docking calculations. Thousands of docking poses were obtained and they were clustered and then rescored by the MM/GBSA method. The selected cluster of representative poses with chiralities *R* and *RS* are presented in Figure 3a. The predicted docking pose for galantamine within AChE active site (using PDB code: 1ODC), as reference compound, was also obtained. It is observed that docked compounds adopted approximately the same position within the active site. The binding modes encompass four X-ray structures with PDB codes 1ODC, 1ZGC, 2CKM, and 2CEK, which exhibit distinct conformations of the side chains of the amino acids Trp279, Phe330, and His440. The THQ moiety was located at the anionic site near the bottom of the active site gorge, which is composed by the amino acids Trp84, Tyr130, Phe330, and Phe331 (magenta in Figure 3), occupying the same space as known compounds such as huprine (PDB code 1E66) [24]. The terminal phenyl group extends to the peripheral anionic site (PAS, orange) at the top of the gorge formed by the amino acids Tyr70, Asp72, Tyr121, Trp279, and Tyr334, resembling the position of alkylene-linked heterodimers of tacrine (PDB code 1ZGC) [25]. This binding orientation is similar to the docking poses obtained for another THQ derivative series reported in a previous work [19]. With respect to galantamine, it established a cation–π interaction with Phe330 and π–π stacking interactions with residue Trp84, and as mentioned above, our compounds occupied the same space within active site but they also extended their isoxazole and phenyl rings towards the PAS going through acyl pocket (residues Phe288 and Phe290) (Figure 3c). Potentially improvement in the interactions with residues composing the PAS site, where there exist some structural differences between AChE and BChE, would allow us to achieve better specificity in a next series of hybrid compounds. It is worth mentioning that it was not possible to obtain analogous docking poses for the four compounds having a chlorine atom at position 8 of the THQs (substituent at R_5_ in Scheme 1), probably due to steric clashes with side chain of residue Tyr334 that may have prevented them to enter the active site in the proper orientation. In general, the predicted docking poses seem to agree with the binding modes reported for known AChE inhibitors, which are placed in well-known pockets described as relevant for inhibitory potency. On the other hand, we also performed molecular docking of compounds 5n and 6aa, as the most actives against AChE and BChE, respectively, within the BChE active site with the aim to understand their potential selectivity on these molecular targets. In Appendix A are shown the most favorable poses for compounds 5n and 6aa within BChE and AChE binding sites. As can be seen, the poses adopted by those compounds resemble the ones obtained for all set of compounds against AChE active site. Specifically, 6aa (IC_50_ BChE (µM) = 3.97), with a docking score (kcal/mol) = −8.203), presented π–π stacking interactions with residues Trp82 and His438; meanwhile, compound 5n (IC_50_ BChE (µM) = 22.00, and a docking score (kcal/mol) = −7.681), only establishes one π–π stacking interaction with Trp82 (see Appendix A). With respect to the high selectivity of compound 6aa (SI = 0.04) against BChE, it is difficult to drawn conclusions using only binding energies obtained with molecular docking. However, it could be argued that structural differences at the PAS site of BChE, which is conformed by less bulky residues like Asn68, Gln119 and Ala277, when compared with PAS site in AChE (residues Tyr70, Tyr121, and Trp279) would be involved in an easier access of compound to gorge in conjunction to better physicochemical properties due to bromide substituent at R_1_ (see Appendix A). In the case of 5n compound´s selectivity against AChE, there is a good correspondence between computational docking calculations and biological experiments. For instance, the IC_50_ AChE (µM) = 4.24 and IC_50_ BChE (µM) = 22.00 values compared well with the corresponding affinities in terms of docking scoring energies that are −10.461 and −7.861 kcal/mol, respectively. The trend in affinity can also be reproduced by docking calculations, which means that compound 5n has more favorable binding energy within AChE binding site (−10.461 kcal/mol) than compound 6aa (−8.127 kcal/mol). This binding energy difference could be due to differences at the PAS site, as was mentioned above, but also to a structural difference at hydrophobic patch, where AChE has a Phe330 and BChE an Ala328. Compound 5n establishes a π–π stacking interaction with Phe330 that cannot be established at BChE active site (see Appendix A). In any case, the differences in residues conforming to the PAS site, the acyl pocket, and to a lesser extent the hydrophobic patch, will be taken into account when designing new hybrid compounds against these cholinesterases. 

Similarly, the predicted free binding energies (∆Gpred) through the MM/GBSA method agreed well with the experimental energies (∆Gexpt, converted from the IC_50_ values), showing a good correlation (R^2^ = 0.83), as evidenced in Figure 4. More importantly, the most and least active compounds were correctly ranked among the 40 compounds.

The specific protein–ligand interactions observed in the representative docking poses are summarized in Table 3. A single ligand is shown within the AChE active site in Figure 3b as an illustrative example. Several π-stacking interactions were identified between the aromatic rings of the THQ and terminal phenyl moieties and the aromatic cavity formed by the side chains of Trp84, Trp279, Phe330, and Tyr334. Particularly, the π-stacking interaction in sandwich conformation with residue Phe330 seems to be an anchor point, as all compounds exhibited such interaction with a very short distance (3.56 ± 0.38 Å), which suggest a strong attraction. Similarly, most ligands also established π-stacking interactions with either Trp84 (in parallel-displaced conformation) or Tyr334 (in T-shaped conformation), or both, with average distances of 4.10 ± 0.18 Å and 4.55 ± 0.62 Å, respectively. These same non-covalent interactions are observed in the X-ray structure of huprine [24]. Some compounds (7) also presented a π-stacking interaction between the terminal phenyl group and the side chain of Trp279, with a distance of 4.92 ± 0.45 Å. Additionally, hydrogen bonds (H-bonds) were identified between the methoxy substituents at the terminal phenyl moiety and the amine of the Trp279 side chain in 11 out of the 40 ligands with an average distance of 3.17 ± 0.28 Å. Furthermore, minor H-bonds were observed with one aromatic hydrogen of Phe330 side chain, either via either the carbonyl oxygen or oxygen of the pyrrolidone moiety, with average distances of 2.68 ± 0.13 Å and 1.99 ± 0.30 Å, respectively. These results suggest that the synthesized ligands interact with AChE primarily by π-stacking and other hydrophobic interactions, whereas H-bonds only help to further stabilize the binding of these inhibitors.

Furthermore, an *in silico* analysis of the ADME pharmacokinetic properties (Appendix A) indicates that there are no significant violations of Lipinski’s rule since the calculated descriptors are within the normal ranges (Log *p* = 3.181–4.881, HB acceptors = 5.500–9.700, and HB = 0.0 donors), except 6o, 6aa, and 6ab compounds that have a molecular weight over 500 atomic mass units (502.396, 512.402, and 542.428 g/mol, respectively). With respect to the predicted qualitative oral absorption, a value of 1, 2, or 3 indicates low, medium, or high absorption, respectively. This prediction was made through the analysis of the adequate values of different descriptors, such as the number of probable metabolic reactions, the number of rotatable bonds, log *p*, solubility and cellular permeability. Most of the compounds had a value of 3, so they could present a good absorption at the oral level. In addition, membrane penetration and distribution of these compounds in organisms are given by parameters such as PSA and solubility, which were analyzed and are within the acceptable range defined for human use. In general, after calculating the *in silico* values of the different properties, it can be inferred that these new THQ–isoxazole/isoxazoline hybrids have characteristics that classify them as potential drugs. Table 4 shows the values of descriptors of the most active compounds. 

## 3. Materials and Methods

### 3.1. Chemistry

All reagents were purchased from Merck (Darmstadt, Germany), J.T. Baker, and Sigma and Aldrich Chemical Co. and they were used without purification. ^1^H NMR and ^13^C NMR were recorded at 400 and 100 MHz, respectively, on a Bruker Ultrashield-400 spectrometer. The chemical shifts (δ) are reported in parts per million (ppm), using CDCl_3_ as solvent with TMS as an internal standard. The coupling constant (*J*) are reported in Hertz (Hz). A FT-IR Bruker Tensor 27 spectrophotometer coupled to Bruker platinum ATR cell was used to obtain IR spectra. Mass spectra were recorded using a Bruker Daltonics ESI-IT Amazon X spectrometer with direct injection, operating in full scan at 300 °C and 4500 V in the capillary. High-resolution mass spectrometry ESI-MS analyses were conducted in a high-resolution hybrid quadrupole (Q) and orthogonal time-of-flight (TOF) mass spectrometer (Waters/Micromass Q-TOF micro, Manchester, UK) with a constant nebulizer temperature of 100 °C. The elemental analysis of the different compounds was performed in ThermoScientific CHNS-O analyzer equipment (Model Flash 2000, Thermo Scientific, Waltham, MA, USA). Melting points (uncorrected) were determined on an Electrothermal IA9100 melting point apparatus. The progress of reaction was monitored by TLC, purification of compounds was performed by column chromatography using silica gel as support. Solvents employed were of analytical grade. 

#### 3.1.1. General Procedure for the Synthesis of Aldoximes

A mixture of appropriate aromatic aldehyde 1 (1 mmol) hydroxylamine hydrochloride (1.1 mmol), and sodium carbonate (1.1 mmol) was dissolved in MeOH and stirred at room temperature for 2 h. The reaction mixture was diluted with water (30 mL) and extracted with ethyl acetate (3 × 15 mL). The organic layer was separated and dried (Na_2_SO_4_). The respective aldoxime 2 was obtained by solvent removal under vacuum.

#### 3.1.2. General Procedure for Isoxazole/Isoxazoline Synthesis

A mixture of *N*-Allyl (3) or *N*-propargyl (4) THQ (1 mmol) and the appropriate aldoxime 2 was dissolved in 8 mL of DCM and stirred for 10 min. Triethylamine (1 mmol) was added and the reaction was cooled to 0 °C and stirred. 8 mL of NaClO was added dropwise. The reaction mixture was stirred 3–4 h at room temperature. Water was added and ethyl acetate was used to extract. The organic layer was separated and dried with NaSO_4_. The solvent was removed and the final product purified by column chromatography using appropriate ethyl acetate and petroleum ether mixture to provide isoxazolines 5 and isoxazoles 6 compounds. NMR spectra of compounds can be found in the Appendix A.

3-phenyl-5-[(4-(2′-oxopyrrolidin-1′-yl)-3,4-dihydroquinolin-1(2H)-yl)-methyl]-4,5 dihydroisoxazole (5a): 74% yield; Yellow oil; IR (ATR): 1664, 1605, 1502, 1357 cm^−1^. ^1^H NMR (400 MHz, CDCl_3_) δ (ppm): 77.63–7.60 (8H, m), 7.40–7.37 (8H, m), 7.18–7.15 (2H, m), 5.41 (2H, q, *J* = 7.2 Hz, 4′-H), 4.80–4.73 (2H, m), 3.54–3.44 (6H, m), 3.32–3.28 (4H, m), 3.24–3.16 (4H, m), 3.05–2.99 (2H, m), 2.45–2.39 (4H, m), 2.01–1.91 (4H, m), 1.49–1.47 (4H, m). ^13^C NMR (100 MHz, CDCl_3_) δ (ppm). Diastereomer α: 174.75, 156.73, 146.21, 138.31, 131.56, 130.18, 129.60, 129.34, 128.78 (2C), 126.88 (2C), 126.50, 126.30, 79.96, 57.88, 57.51, 48.32, 42.43, 38.53, 31.41, 17.96, 16.48. Diastereomer β: 174.75, 156.68, 146.21, 138.07, 131.56, 130.16, 129,61, 129.30, 128.78 (2C), 126.88 (2C), 126.48, 126.27, 79,96, 57.86, 57.50, 48.28, 42.41, 38.53, 31.41, 17.96, 16.42. ESI-MS (m/z): 376 [M + H]^+^, 398 [M + Na]^+^, 414 [M + K]^+^, 763 [2M + Na]^+^, 291 [(M + H)-C_4_H_6_NO]^+^, 160 [(M + H)-C_13_H_15_N_2_O]^+^. Anal. Calcd. for C_23_H_25_N_3_O_2_: C, 73.57; H, 6.71; N, 11.19%. Found: C, 72.45; H, 6.59; N, 10.89%.

3-(4-methoxyphenyl)-5-[4-((2′-oxopyrrolidin-1′-yl)-3,4-dihydroquinolin-1(2H)-yl)-methyl]-4,5-dihydroisoxazole (5b): 64% yield; Yellow oil; IR (ATR): 1668, 1608, 1512, 1355 cm^−1^. ^1^H NMR (400 MHz, CDCl_3_) δ (ppm): 7.63–7.58 (4H, m), 7.24–7.20 (2H, m), 7.04 (2H, m), 6.93–6.87 (4H, m), 6.76 (2H, m), 6.58 (2H, m), 5.34 (2H, q, *J* = 7.5 Hz), 5.21–5.11 (2H, m), 3.82 (3H, sα), 3.81 (3H, sβ), 3.60–3.23 (14H, m), 3.10–3.01 (2H, m), 2.51–2.42 (4H, m), 2.08–1.97 (4H, m), 1.58–1.52 (4H, m). ^13^C NMR (100 MHz, CDCl_3_) δ (ppm). Diastereomer α: 174.74, 156.94, 145.45, 130.82, 129.86, 128.04, 128.32 (2C), 127.49, 126.50, 122.12, 114.27 (2C), 113.19, 81.87, 57.29, 55.45, 48.22, 47.0, 43.37, 38.78, 31.38, 18.17, 16.88. Diastereomer β: 174.67, 156.58, 144.93, 130.60, 129.77, 128.01, 128.32 (2C), 127.45, 126.16, 121.98, 114.27 (2C), 113.02, 81.70, 56.79, 55.45, 47.33, 46.3, 42.83, 38.11, 31.17, 17.99, 16.75. ESI-MS (m/z): 406 [M + H]^+^, 428 [M + Na]^+^, 444 [M + K]^+^, 833 [2M + Na]^+^, 321 [(M + H)-C_4_H_6_NO]^+^, 190 [(M + H)-C_13_H_15_N_2_O]^+^. Anal. Calcd. for C_24_H_27_N_3_O_3_: C, 71.09; H, 6.71; N, 10.36%. Found: C, 72.08; H, 6.62; N, 10.08%.

3-(3,4-dimethoxyphenyl)-5-[4-((2′-oxopyrrolidin-1′-yl)-3,4-dihydroquinolin-1(2H)-yl)-methyl]-4,5-dihydroisoxazole (5c): 65% yield; Yellow oil; IR (ATR): 1666, 1602, 1514, 1366 cm^−1^. ^1^H NMR (400 MHz, CDCl_3_) δ (ppm): 7.37 (1H, sα), 7.35 (1H, sβ), 7.23–7.14 (2H, m), 7.02–6.97 (2H, m), 6.85–6.91 (4H, m), 6.75–6.69 (4H, m), 5.40 (2H, q, *J* = 7.4 Hz), 5.08–4.99 (2H, m), 3.90 (6H, sα), 3.89 (6H, sβ), 3.56–3.37 (10H, m), 3.25–3.01 (6H, m), 2.42–2.38 (4H, m), 2.03–1.90 (4H, m), 1.47–1.44 (4H, m). ^13^C NMR (100 MHz, CDCl_3_) δ (ppm). Diastereomer α: 174.71, 156.86, 151.00, 149.19, 147.20, 129.68, 128.29, 128.22, 122.38, 120.50, 118.60, 113.17, 110.53, 108.66, 79.76, 56.02 (2C), 48.54, 48.27, 47.18, 42.27, 38.25, 31.68, 17.94, 16.49. Diastereomer β: 174.42, 156.47, 150.83, 149.12, 147.20, 129.35, 128.29, 128.35, 122.10, 120.50, 118.60, 113.17, 110.45, 108.60, 79.42, 56.02 (2C), 48.54, 48.17, 47.18, 42.21, 38.25, 31.40, 17.90, 16.38. ESI-MS (m/z): 436 [M + H]^+^, 458 [M + Na]^+^, 474 [M + K]^+^, 893 [2M + Na]^+^, 351 [(M + H)-C_4_H_6_NO]^+^, 220 [(M + H)-C_13_H_15_N_2_O]+. Anal. Calcd. for C_25_H_29_N_3_O_4_: C, 68.95; H, 6.71; N, 9.65%. Found: C, 68.39; H, 6.59; N, 9.89%.

3-(3,4,5-trimethoxyphenyl)-5-[4-((2′-oxopyrrolidin-1′-yl)-3,4-dihydroquinolin-1(2H)-yl)-methyl]-4,5-dihydroisoxazole (5d): 65% yield; Red oil; IR (ATR): 1666, 1606, 1514, 1357 cm^−1^. ^1^H NMR (400 MHz, CDCl_3_) δ (ppm): 7.36–7.30 (2H, m), 7.14 (2H, m), 7.01–6.99 (2H, m), 6.88 (2H, sα), 6.86 (2H, sβ), 6.63 (2H, m), 5.37 (2H, q, *J* = 6.5 Hz, 4′-H), 4.94–4.80 (2H, m, 5-H), 3.89 (6H, s), 3.85 (6H, s), 3.84 (6H, s), 3.60–3.40 (8H, m), 3.39–3.23 (6H, m), 3.05–2.97 (2H, m), 2.46–2.35 (4H, m), 2.16–1.99 (4H, m), 1.53–1.49 (4H, m). ^13^C NMR (100 MHz, CDCl_3_) δ (ppm). Diastereomer α: 175.48, 155.32, 153.50 (2C), 144.87, 140.12, 128.05, 127.12, 124.94, 121.65, 121.45, 112.58, 104.19 (2C), 79.88, 61.32, 56.55 (2C), 54.74, 48.90, 47.88, 43.12, 38.42, 31.19, 18.34, 16.94. Diastereomer β: 175.39, 155.32, 153.42 (2C), 144.75, 139.95, 127.94, 127.08, 124.87, 121.58, 121.38, 112.45, 104.12 (2C), 79.64, 61.29, 56.50 (2C), 54.70, 48.75, 47.80, 42.95, 38.40, 31.12, 18.29, 16.78. ESI-MS (m/z): 466 [M + H]^+^, 488 [M + Na]^+^, 504 [M + K]^+^, 953 [2M + Na]^+^, 381 [(M + H)-C_4_H_6_NO]^+^, 250 [(M + H)-C_13_H_15_N_2_O]^+^. Anal. Calcd. for C_26_H_31_N_3_O_5_: C, 68.95; H, 6.71; N, 9.65%. Found: C, 68.45; H, 6.65; N, 9.82%.

5-[(6-methyl-4-(2′-oxopyrrolidin-1′-yl)-3,4-dihydroquinolin-1(2H)-yl)-methyl]-3-phenyl-4,5-dihydroisoxazole (5e): 65% yield; Orange oil; IR (ATR): 1664, 1600, 1496, 1357 cm^−1^. ^1^H NMR (400 MHz, CDCl_3_) δ (ppm): 7.68–764 (6H, m), 7.42–7.39 (4H, m), 6.93 (1H, ddα, *J* = 8.4, 2.0 Hz), 6.91 (1H, ddβ, *J* = 8.4, 2.0 Hz), 6.72 (1H, dα, *J* = 2.0 Hz), 6.67 (1H, dβ, *J* = 2.0 Hz), 6.54 (2H, d, *J* = 8.4 Hz), 5.37(1H, ddα, *J* = 9.4, 5.8 Hz), 5.31(1H, ddβ, *J* = 9.4, 5.8 Hz), 5.09–5.01 (2H, m), 3.56–3.37 (10H, m), 3.25–3.01 (6H, m), 2.51–2.46 (4H, m), 2.20 (3H, sα), 2.19 (3H, sβ), 2.14–2.05 (4H, m), 1.98–1.90 (4H, m). ^13^C NMR (100 MHz, CDCl_3_) δ (ppm). Diastereomer α: 175.62, 156.71, 144.02, 130.35, 129.46, 129.38, 128.87 (2C), 128.84, 126.77 (2C), 126.05, 119.73, 111.47, 79.86, 55.35, 48.96, 48.07, 44.39, 38.23, 31.63, 26.97, 20.40, 18.43. Diastereomer β: 175.51, 156.70, 143.50, 130.28, 129.33, 128.99, 128.84 (2C), 128.51, 126.77 (2C), 125.90, 119.51, 111.27, 79.43, 54.96, 48.79, 47.95, 43.67, 37.92, 31.58, 26.71, 20.38, 18.34. ESI-MS (m/z): 390 [M + H]^+^, 412 [M + Na]^+^, 428 [M + K]+, 801 [2M + Na]^+^, 305 [(M + H)-C_4_H_6_NO]^+^, 243 [(M + H)-C_9_H_8_NO]^+^, 160 [(M + H)-C_14_H_17_N_2_O]^+^. Anal. Calcd. for C_24_H_27_N_3_O_2_: C, 74.01; H, 6.99; N, 10.79%. Found: C, 73.76; H, 6.87; N, 10.54%.

3-(4-methoxyphenyl)-5-[(6-methyl-4-(2′-oxopyrrolidin-1′-yl)-3,4-dihydroquinolin-1(2H)-yl)-methyl]-4,5-dihydroisoxazole (5f): 70% yield; Orange oil; IR (ATR): 1676, 1606, 1514, 1359 cm^−1^. ^1^H NMR (400 MHz, CDCl_3_) δ (ppm): 7.62 (2H, dα, *J* = 8.9 Hz), 7.61 (2H, dβ, *J* = 8.9 Hz), 6.93 (2H, dα, *J* = 8.8 Hz), 6.91 (2H, dβ, *J* = 8.9 Hz), 6.72 (2H, dd, *J* = 8.6, 2.7 Hz), 6.63(1H, dα, *J* = 2.7 Hz), 6.61 (1H, dβ, *J* = 2.7 Hz), 6.50 (1H, dα, *J* = 8.6 Hz), 6.49 (1H, dβ, *J* = 8.6 Hz), 5.39 (1H, ddα, *J* = 9.2, 5.6 Hz), 5.34 (1H, ddβ, *J* = 9.8, 6.5 Hz), 5.08–4.99 (2H, m), 3.89 (3H, sα), 3.83 (3H, sβ), 3.58–3.35 (10H, m), 3.24–3.01 (6H, m), 2.51–2.45 (4H, m), 2.22 (3H, sα), 2.21 (3H, sβ), 2.11–2.03 (4H, m), 2.01–1.92 (4H, m). ^13^C NMR (100 MHz, CDCl_3_) δ (ppm). Diastereomer α: 174.78, 155.10, 150.90, 144.195, 129.44, 128.94 (2C), 128.44, 127.37, 122.12, 121.16, 114.41 (2C), 113.88, 82.01, 57.12, 55.52, 47.20, 46.77, 43.55, 38.72, 31.34, 27.80, 19.30, 18.37. Diastereomer β: 174.56, 154.88, 150.90, 143.75, 129.23, 128.94 (2C), 128.34, 126.88, 122.07, 121.16, 114.18 (2C), 113.88, 81.79, 56.46, 55.47, 47.20, 46.77, 42.61, 37.68, 31.18, 27.80, 19.30, 17.75. ESI-MS (m/z): 420 [M + H]^+^, 442 [M + Na]^+^, 458 [M + K]^+^, 861 [2M + Na]^+^, 335 [(M + H)-C_4_H_6_NO]^+^, 243 [(M + H)-C_10_H_10_NO_2_]^+^, 190 [(M + H)-C_14_H_17_N_2_O]^+^. Anal. Calcd. for C_25_H_29_N_3_O_3_: C, 71.57; H, 6.97; N, 10.02%. Found: C, 72.06; H, 6.83; N, 10.29%.

3-(3,4-dimethoxyphenyl)-5-[(6-methyl-4-(2′-oxopyrrolidin-1′-yl)-3,4-dihydroquinolin-1(2H)-yl)-methyl]-4,5-dihydroisoxazole (5g): 63% yield; Orange oil; IR (ATR): 1666, 1598, 1508, 1365 cm^−1^. ^1^H NMR (400 MHz, CDCl_3_) δ (ppm): 7.40 (2H, s), 7.07–7.00 (4H, m), 6.92 (1H, ddα, *J* = 8.6, 2.0 Hz), 6.90 (1H, ddβ, *J* = 8.6, 2.0 Hz), 6.70 (1H, dα, *J* = 2.0Hz), 6.66 (1H, dβ, *J* = 2.0 Hz), 6.53 (2H, d, *J* = 8.6 Hz), 5.34 (1H, ddα, *J* = 9.5, 5.8 Hz), 5.30 (1H, ddβ, *J* = 7.4, 5.4 Hz), 5.04–4.99 (2H, m, 5-H), 3.90 (6H, sα), 3.89 (6H, sβ), 3.50–3.38 (10H, m), 3.24–3.03 (6H, m), 2.50–2.47 (4H, m), 2.19 (3H, sα), 2.18 (3H, sβ), 2.11–2.04 (4H, m), 1.99–1.94 (4H, m). ^13^C NMR (100 MHz, CDCl_3_) δ (ppm). Diastereomer α: 175.68, 156.47, 150.98, 149.20, 143.98, 129.32, 128.93, 127.48, 126.00, 120.46, 119.68, 111.51, 110.51, 108.61, 79.65, 56.09 (2C), 48.90, 48.10, 47.19, 43.70, 38.09, 31.61, 26.94, 20.38, 18.34. Diastereomer β: 175.56, 156.47, 150.93, 149.18, 143.47, 129.19, 128.44, 127.35, 125.84, 120.43, 119.45, 111.27, 110.47, 108.61, 79.26, 56.02 (2C), 48.77, 47.99, 47.14, 43.54, 37.84, 31.56, 26.68, 20.36, 18.30. ESI-MS (m/z): 450 [M + H]^+^, 472 [M + Na]^+^, 488 [M + K]^+^, 921 [2M + Na]^+^, 365 [(M + H)-C_4_H_6_NO]^+^, 243 [(M + H)-C_11_H_12_NO_3_]^+^, 220 [(M + H)-C_14_H_17_N_2_O]^+^. Anal. Calcd. for C_26_H_31_N_3_O_4_: C, 69.47; H, 6.95; N, 9.35%. Found: C, 70.12; H, 6.79; N, 9.61%. 

5-[(6-methyl-4-(2′-oxopyrrolidin-1′-yl)-3,4-dihydroquinolin-1(2H)-yl)-methyl]-3-(3,4,5-trimethoxyphenyl)-4,5-dihydroisoxazole (5h): 68% yield; Orange oil; IR (ATR): 1664, 1598, 1508, 1369 cm^−1^. ^1^H NMR (400 MHz, CDCl_3_) δ (ppm): 6.92 (2H, sα), 6.90 (2H, sβ), 6.73 (1H, dα, *J* = 2.6 Hz), 6.71 (1H, dβ, *J* = 2.6 Hz), 6.64 (1H, ddα, *J* = 8.8, 2.6 Hz), 6.63 (1H, ddβ, *J* = 8.8, 2.6 Hz), 6.55 (2H, dd, *J* = 8.8 Hz), 5.39 (1H, ddα, *J* = 9.6, 6.0 Hz), 5.33 (1H, ddβ, *J* = 7.4, 5.6 Hz), 5.05–4.98 (2H, m, 5-H), 3.90 (6H, s), 3.89 (6H, s), 3.87 (6H, s), 3.58–3.35 (10H, m), 3.23–3.00 (6H, m), 2.53–2.45 (4H, m), 2.23 (3H, sα), 2.21 (3H, sβ), 2.14–2.06 (4H, m), 2.01–1.94 (4H, m). ^13^C NMR (100 MHz, CDCl_3_) δ (ppm). Diastereomer α: 175.88, 155.40, 153.56 (2C), 144.10, 139.40, 129.70, 128.74, 125.65, 124.22, 121.41, 111.54, 104.12 (2C), 79.94, 61.08, 56.43 (2C), 55.77, 47.63, 47.20, 43.60, 38.02, 31.58, 26.46, 20.65, 18.41. Diastereomer β: 175.62, 155.40, 153.49 (2C), 143.69, 139.40, 129.70, 128.74, 125.65, 124.22, 121.41, 111.54, 104.06 (2C), 78.22, 61.04, 56.38 (2C), 54.31, 47.58, 46.75, 42.88, 37.52, 31.23, 26.46, 20.65, 17.85. ESI-MS (m/z): 480 [M + H]^+^, 502 [M + Na]^+^, 518 [M + K]^+^, 981 [2M + Na]^+^, 395 [(M + H)-C_4_H_6_NO]^+^, 250 [(M + H)-C_14_H_17_N_2_O]^+^. Anal. Calcd. for C_27_H_33_N_3_O_5_: C, 67.62; H, 6.94; N, 8.76%. Found: C, 68.23; H, 6.83; N, 8.54%.

5-[(6-methoxy-4-(2′-oxopyrrolidin-1′-yl)-3,4-dihydroquinolin-1(2H)-yl)-methyl]-3-phenyl-4,5-dihydroisoxazole (5i): 77% yield; Red oil; IR (ATR): 1668, 1606, 1502, 1357 cm^−1^. ^1^H NMR (400 MHz, CDCl_3_) δ (ppm): 7.67–7.64 (4H, m), 7.41–7.37 (6H, m), 6.73 (1H, ddα, *J* = 8.7, 2.6 Hz), 6.71 (1H, ddβ, *J* = 8.7, 2.6 Hz), 6.59 (2H, d, *J* = 8.7 Hz), 6.51 (1H, dα, *J* = 2.6 Hz), 6.46 (1H, dβ, *J* = 2.6 Hz), 5.37 (1H, ddα, *J* = 9.6, 6.0 Hz), 5.33 (1H, ddβ, *J* = 8.4, 5.6 Hz), 5.06–4.99 (2H, m), 3.70 (3H, sα), 3.69 (3H, sβ), 3.53–3.35 (10H, m), 3.25–3.05 (6H, m), 2.49–2.44 (4H, m), 2.17–2.05 (4H, m), 1.98- 1.90 (4H, m). ^13^C NMR (100 MHz, CDCl_3_) δ (ppm). Diastereomer α: 175.48, 156.65, 151.37, 140.30, 130.22, 129.33, 128.78 (2C), 126.69 (2C), 120.95, 114.16, 113.57, 112.55, 79.47, 55.86, 55.52, 48.84, 48.09, 43.41, 37.93, 31.44, 26.60, 18.28. Diastereomer β: 175.48, 156.65, 151.37, 140.30, 130.22, 129.33, 128.78 (2C), 126.69 (2C), 120.95, 114.16, 113.57, 112.55, 79.47, 55.86, 55.52, 48.84, 48.09, 43.41, 37.93, 31.44, 26.60, 18.28. ESI-MS (m/z): 406 [M + H]^+^, 428 [M + Na]^+^, 444 [M + K]^+^, 833 [2M + Na]^+^, 321 [(M + H)-C_4_H_6_NO]^+^, 259 [(M + H)-C_9_H_8_NO]^+^, 160 [(M + H)-C_14_H_17_N_2_O_2_]^+^. Anal. Calcd. for C_24_H_27_N_3_O_3_: C, 71.09; H, 6.71; N, 10.36%. Found: C, 70.78; H, 6.67; N, 10.12%.

3-(4-methoxyphenyl)-5-[(6-methoxy-4-(2′-oxopyrrolidin-1′-yl)-3,4-dihydroquinolin-1(2H)-yl)-methyl]-4,5-dihydroisoxazole (5j): 70% yield; Red oil; IR (ATR): 1664, 1602, 1508, 1369 cm^−1^. ^1^H NMR (400 MHz, CDCl_3_) δ (ppm): 7.62 (4H, d, *J* = 8.8 Hz), 6.91 (2H, dα, *J* = 8.8 Hz), 6.90 (2H, dβ, *J* = 8.8 Hz), 6.72 (1H, ddα, *J* = 8.9, 2.6 Hz), 6.70 (1H, ddβ, *J* = 8.9, 2.6 Hz), 6.59 (1H, dα, *J* = 2.6 Hz), 6.57 (1H, dβ, *J* = 2.6 Hz), 6.50 (1H, dα, *J* = 8.9 Hz), 6.49 (1H, dβ, *J* = 8.9 Hz), 5.39 (1H, ddα, *J* = 9.6, 6.0 Hz), 5.33 (1H, ddβ, *J* = 7.4, 5.6 Hz), 5.05–4.97 (2H, m, 5-H), 3.83 (3H, sα), 3.82 (3H, sβ), 3.70 (3H, sα), 3.69 (3H, sβ), 3.50–3.38 (10H, m), 3.22–3.09 (6H, m), 2.49–2.46 (4H, m), 2.16–2.07 (4H, m), 1.98–1.94 (4H, m). ^13^C NMR (100 MHz, CDCl_3_) δ (ppm). Diastereomer α: 175.64, 156.63, 155.08, 151.49, 140.84, 128.32 (2C), 122.11, 121.25, 115.99, 114.21 (2C), 112.81, 111.97, 79.52, 55.94, 55.61, 55.22, 48.94, 47.39, 43.40, 38.50, 31.51, 26.89, 18.42. Diastereomer β: 175.54, 156.28, 154.93, 151.37, 140.32, 128.30 (2C), 122.02, 120.96, 115.76, 113.63 (2C), 112.59, 111.64, 79.19, 55.79, 55.46, 54.13, 48.27, 46.94, 42.79, 38.26, 31.29, 26.66, 18.34. ESI-MS (m/z): 436 [M + H]^+^, 458 [M + Na]^+^, 474 [M + K]^+^, 893 [2M + Na]^+^, 351 [(M + H)-C4H6NO]^+^, 259 [(M + H)-C_10_H_10_NO_2_]^+^, 190 [(M + H)-C_14_H_17_N_2_O_2_]+. Anal. Calcd. for C_25_H_29_N_3_O_4_: C, 68.95; H, 6.71; N, 9.65%. Found: C, 68.78; H, 6.54; N, 9.41%.

3-(3,4-dimethoxyphenyl)-5-[(6-methoxy-4-(2′-oxopyrrolidin-1′-yl)-3,4-dihydroquinolin-1(2H)-yl)-methyl]-4,5-dihydroisoxazole (5k): 76% yield; Red oil; IR (ATR): 1666, 1602, 1502, 1357 cm^−1^. ^1^H NMR (400 MHz, CDCl_3_) δ (ppm): 7.39 (2H, d, *J* = 2.0 Hz), 7.02 (1H, ddα, *J* =8.3, 2.0 Hz), 7.00 (1H, ddβ, *J* = 8.3, 2.0 Hz), 6.85 (2H, dα, *J* = 8.3 Hz), 6.83 (2H, dβ, *J* = 8.3 Hz), 6.73 (1H, ddα, *J* = 9.0, 2.8 Hz), 6.71 (1H, ddβ, *J* = 9.0, 2.8 Hz), 6.58 (2H, d, *J* = 9.0 Hz), 6.50 (1H, d, *J* = 2.8 Hz), 6.46 (1H, dβ, *J* = 2.8 Hz), 5.39 (1H, dd, *J* = 9.6, 6.0 Hz), 5.33 (1H, dd, *J* = 8.4, 5.6 Hz),5.05–4.97 (2H, m), 3.90 (12H, s), 3.70 (3H, sα), 3.69 (3H, sβ), 3.54–3.36 (10H, m), 3.27–3.04 (6H, m), 2.50–2.45 (4H, m), 2.17–2.05 (4H, m), 1.99–1.92 (4H, m). ^13^C NMR (100 MHz, CDCl_3_) δ (ppm). Diastereomer α: 175.68, 156.49, 151.53, 151.00, 149.22, 140.81, 122.26, 121.31, 120.47, 114.46, 113.97, 112.84, 110.53, 108.63, 79.67, 56.07, 55.95, 55.86, 49.06, 48.29, 44.18, 38.40, 31.56, 26.89, 21.18, 18.43. Diastereomer β: 175.58, 156.48, 151.42, 150.96, 149.22, 140.35, 122.19, 121.00, 120.44, 114.23, 113.60, 112.60, 110.49, 108.62, 79.37, 56.03, 55.94, 55.64, 48.92, 48.18, 43.51, 38.18, 31.52, 26.66, 20.91, 18.37. ESI-MS (m/z): 466 [M + H]^+^, 488 [M + Na]^+^, 504 [M + K]^+^, 953 [2M + Na]^+^, 381 [(M + H)-C_4_H_6_NO]^+^, 259 [(M + H) C_11_H_12_NO_3_]^+^, 220 [(M + H)-C_14_H_17_N_2_O_2_]^+^. Anal. Calcd. for C_26_H_31_N_3_O_5_: C, 67.08; H, 6.71; N, 9.03%. Found: C, 67.41; H, 6.59; N, 9.29%.

5-[(6-methoxy-4-(2′-oxopyrrolidin-1′-yl)-3,4-dihydroquinolin-1(2H)-yl)-methyl]-3-(3,4,5-trimethoxyphenyl)-4,5-dihydroisoxazole (5l): 68% yield; Red oil; IR (ATR): 1670, 1597, 1504, 1369 cm^−1^. ^1^H NMR (400 MHz, CDCl_3_) δ (ppm): 6.88 (2H, sα), 6.86 (2H, sβ),6.72 (1H, ddα, *J* = 8.7, 2.8 Hz), 6.71 (1H, ddβ, *J* = 8.7, 2.8 Hz), 6.59 (1H, dα, *J* = 8.7 Hz), 6.57 (1H, dβ, *J* = 8.7 Hz), 6.50 (1H, dα, *J* = 2.8 Hz), 6.47 (1H, dβ, *J* = 2.8 Hz), 5.40 (1H, ddα, *J* = 9.6, 5.9 Hz), 5.33 (1H, ddβ, *J* = 8.6, 5.6 Hz), 5.07–5.00 (2H, m), 3.87 (12H, s), 3.86 (3H, sα), 3.85 (3H, sβ), 3.70 (3H, sα), 3.69 (3H, sβ), 3.54–3.37(10H, m), 3.25–3.05 (6H, m), 2.50–2.45 (4H, m), 2.17–2.07 (4H, m), 2.01–1.97 (4H, m). ^13^C NMR (100 MHz, CDCl_3_) δ (ppm). Diastereomer α: 175.69, 156.59, 153.43 (2C), 151.57, 140.77, 139.96, 124.91, 121.40, 114.43, 113.95, 112.87, 104.02 (2C), 79.94, 61.08, 56.38 (2C), 55.95, 55.84, 49.06, 48.27, 44.15, 38.35, 31.58, 26.87, 18.44. Diastereomer β: 175.57, 156.59, 153.39 (2C), 151.47, 140.33, 139.90, 124.85, 121.10, 114.20, 113.56, 112.58, 104.02 (2C), 79.68, 61.08, 56.38 (2C), 55.92, 55.62, 48.93, 48.18, 43.46, 38.14, 31.48, 26.65, 18.35. ESI-MS (m/z): 496 [M + H]^+^, 518 [M + Na]^+^, 534 [M + K]^+^, 1013 [2M + Na]^+^, 411 [(M + H)-C_4_H_6_NO]^+^, 259 [(M + H)-C_12_H_14_NO_4_]^+^, 250 [(M + H)-C_14_H_17_N_2_O_2_]^+^. Anal. Calcd. for C_27_H_33_N_3_O_6_: C, 65.44; H, 6.71; N, 8.48%. Found: C, 65.21; H, 6.60; N, 8.29%.

5-[(6-chloro-4-(2′-oxopyrrolidin-1′-yl)-3,4-dihydroquinolin-1(2H)-yl)-methyl]-3-phenyl-4,5-dihydroisoxazole (5m): 74% yield; Yellow oil; IR (ATR): 1676, 1597, 1497, 1356 cm^−1^. ^1^H NMR (400 MHz, CDCl_3_) δ (ppm): 7.67–7.62 (4H, m), 7.42–7.38 (6H, m), 7.04 (1H, ddα, *J* = 8.9, 2.6 Hz), 7.03 (1H, ddβ, *J* = 8.9, 2.6 Hz), 6.83 (1H, dα, *J* = 2.6 Hz), 6.79 (1H, dβ, *J* = 2.6 Hz), 6.55 (1H, dα, *J* = 8.9 Hz), 6.53 (1H, dβ, *J* = 8.9 Hz), 5.34 (1H, ddα, *J* = 9.8, 5.7 Hz), 5.29 (1H, ddβ, *J* = 8.8, 5.4 Hz), 5.05–4.98 (2H, m), 3.58–3.37 (10H, m), 3.27–3.01 (6H, m), 2.52–2.44 (4H, m), 2.16–2.04 (4H, m), 1.98–1.91 (4H, m). ^13^C NMR (100 MHz, CDCl_3_) δ (ppm). Diastereomer α: 175.64, 156.67, 144.64, 130.39, 129.22, 128.84 (2C), 128.47, 127.57, 126.69 (2C), 121.50, 121.32, 112.51, 79.45, 55.17, 48.80, 47.85, 43.90, 38.13, 31.36, 26.28, 18.30. Diastereomer β: 175.57, 156.67, 144.15, 130.33, 129.16, 128.82 (2C), 128.38, 127.22, 126.69 (2C), 121.36, 121.12, 112.23, 79.07, 54.87, 48.78, 47.77, 43.32, 37.85, 31.31, 26.14, 18.23. ESI-MS (m/z): 410 [M + H]^+^, 432 [M + Na]^+^, 448 [M + K]^+^, 841 [2M + Na]^+^, 325 [(M + H)-C_4_H_6_NO]^+^, 160 [(M + H)-C_13_H_14_ClN_2_O]^+^. Anal. Calcd. for C_23_H_24_ClN_3_O_2_: C, 67.39; H, 5.90; N, 10.25%. Found: C, 66.98; H, 5.84; N, 9.99%.

5-[(6-chloro-4-(2′-oxopyrrolidin-1′-yl)-3,4-dihydroquinolin-1(2H)-yl)-methyl]-3-(4-methoxyphenyl)-4,5-dihydroisoxazole (5n): 64% yield; Yellow oil; IR (ATR): 1673, 1604, 1497, 1356 cm^−1^. ^1^H NMR (400 MHz, CDCl_3_) δ (ppm): 7.59 (2H, dα, *J* = 9.0 Hz), 7.58 (2H, dβ, *J* = 9.0 Hz), 7.04 (1H, ddα, *J* = 8.8, 2.6 Hz), 7.03 (1H, ddβ, *J* = 8.8, 2.6 Hz), 6.92 (2H, dα, *J* = 9.0 Hz), 6.90 (2H, dβ, *J* = 9.0 Hz), 6.83 (1H, dα, *J* = 2.6 Hz), 6.79(1H, dβ, *J* = 2.6 Hz), 6.55 (1H, dα, *J* = 8.8 Hz), 6.53 (1H, dβ, *J* = 8.8 Hz), 5.35 (1H, ddα, *J* = 9.7, 5.6 Hz), 5.30 (1H, ddβ, *J* = 8.7, 5.2 Hz), 5.02–4.95 (2H, m), 3.83 (3H, sα), 3.82 (3H, sβ), 3.58–3.33 (10H, m), 3.27–3.00 (6H, m), 2.53–2.45 (4H, m), 2.16–2.04 (4H, m), 2.00–1.94 (4H, m). ^13^C NMR (100 MHz, CDCl_3_) δ (ppm). Diastereomer α: 175.63, 156.22, 144.80, 128.53, 128.50 (2C), 128.45, 127.57, 121.76, 121.45, 121.25, 114.22 (2C), 112.51, 79.11, 55.49, 55.23, 48.85, 47.95, 43.99, 38.46, 31.46, 26.37, 18.39. Diastereomer β: 175.56, 156.22, 144.17, 128.53, 128.50 (2C), 128.34, 127.22, 121.69, 121.34, 121.09, 114.17 (2C), 112.19, 78.73, 55.49, 54.95, 48.85, 47.85, 43.42, 38.21, 31.40, 26.20, 18.32. ESI-MS (m/z): 440 [M + H]^+^, 462 [M + Na]^+^, 478 [M + K]^+^, 901 [2M + Na]^+^, 355 [(M + H)-C_4_H_6_NO]^+^, 190 [(M + H)-C_13_H_14_ClN_2_O]^+^. Anal. Calcd. for C_24_H_26_ClN_3_O_3_: C, 65.52; H, 5.96; N, 9.55%. Found: C, 65.98; H, 5.88; N, 9.41%. 

3-(4,5-dimethoxyphenyl)-5-[(6-chloro-4-(2′-oxopyrrolidin-1′-yl)-3,4-dihydroquinolin-1(2H)-yl)-methyl]-4,5-dihydroisoxazole (5o): 75% yield; Yellow oil; IR (ATR): 1674, 1599, 1499, 1369 cm^−1^. ^1^H NMR (400 MHz, CDCl_3_) δ (ppm): 7.38 (2H, s), 7.07–6.98 (4H, m), 6.85 (1H, ddα, *J* = 8.6, 2.6 Hz), 6.83 (1H, ddβ, *J* = 8.6, 2.6 Hz), 6.83 (1H, dα, *J* = 2.6 Hz), 6.79 (1H, dβ, *J* = 2.6 Hz), 6.55 (1H, dα, *J* = 8.6 Hz), 6.53 (1H, dβ, *J* = 8.6 Hz), 5.35 (1H, ddα, *J* = 9.7, 5.6 Hz), 5.29 (1H, ddβ, *J* = 8.8, 5.3 Hz), 5.03–4.96 (2H, m), 3.90 (6H, sα), 3.89 (6H, sβ), 3.57–3.39 (10H, m), 3.24–3.02 (6H, m), 2.53–2.45 (4H, m), 2.19–2.05 (4H, m), 1.99–1.94 (4H, m). ^13^C NMR (100 MHz, CDCl_3_) δ (ppm). Diastereomer α: 175.74, 156.49, 151.10, 149.26, 144.70, 128.54, 127.65, 122.09, 121.56, 121.38, 120.50, 112.60, 110.56, 108.62, 79.33, 56.07 (2C), 55.24, 48.86, 47.96, 43.99, 38.34, 31.44, 26.36, 18.38. Diastereomer β: 175.66, 156.49, 151.06, 149.26, 144.21, 128.45, 127.28, 122.02, 121.43, 121.19, 120.47, 112.28, 110.53, 108.62, 78.98, 56.04 (2C), 54.98, 48.86, 47.87, 43.43, 38.10, 31.39, 26.21, 18.33. ESI-MS (m/z): 470 [M + H]^+^, 492 [M + Na]^+^, 508 [M + K]^+^, 961 [2M + Na]^+^, 385 [(M + H)-C_4_H_6_NO]^+^, 220 [(M + H)-C_13_H_14_ClN_2_O]^+^. Anal. Calcd. for C_25_H_28_ClN_3_O_4_: C, 63.89; H, 6.01; N, 8.94%. Found: C, 64.18; H, 6.13; N, 9.11%.

5-[(6-chloro-4-(2′-oxopyrrolidin-1′-yl)-3,4-dihydroquinolin-1(2H)-yl)-methyl]-3-(3,4,5-trimethoxyphenyl)-4,5-dihydroisoxazole (5p): 75% yield; Yellow oil; IR (ATR): 1674, 1597, 1501, 1370 cm^−1^. ^1^H NMR (400 MHz, CDCl_3_) δ (ppm): 7.05 (1H, ddα, *J* = 8.9, 2.6 Hz), 7.03 (1H, ddβ, *J* = 8.9, 2.6 Hz), 6.88 (2H, sα), 6.86 (2H, sβ), 6.83 (1H, dα, *J* = 2.6 Hz), 6.79 (1H, dβ, *J* = 2.6 Hz), 6.55 (2H, d, *J* = 8.9 Hz), 5.37 (1H, ddα, *J* = 9.8, 5.4 Hz), 5.30 (1H, ddβ, *J* = 9.2, 5.2 Hz), 5.07–4.99 (2H, m), 3.88 (6H, s), 3.87 (6H, s), 3.86 (6H, s), 3.57–3.40 (10H, m), 3.22–3.02 (6H, m), 2.52–2.46 (4H, m), 2.19–2.04 (4H, m), 2.01–1.97 (4H, m). ^13^C NMR (100 MHz, CDCl_3_) δ (ppm). Diastereomer α: 175.76, 156.60, 153.49 (2C), 144.66, 140.11, 128.55, 127.64, 124.75, 121.64, 121.47, 112.61, 104.07 (2C), 79.64, 61.13, 56.39 (2C), 55.26, 48.93, 47.97, 43.96, 38.31, 31.46, 26.36, 18.40. Diastereomer β: 175.65, 156.60, 153.49 (2C), 144.16, 139.99, 128.44, 127.25, 124.67, 121.55, 121.29, 112.27, 104.07 (2C), 79.33, 60.54, 54.39 (2C), 54.95, 48.87, 47.85, 43.37, 38.09, 31.37, 26.20, 18.32. ESI-MS (m/z): 500 [M + H]^+^, 522 [M + Na]^+^, 538 [M + K]^+^, 1021 [2M + Na]^+^, 415 [(M + H)-C_4_H_6_NO]^+^, 250 [(M + H)-C_13_H_14_ClN_2_O]^+^. Anal. Calcd. for C_26_H_30_ClN_3_O_5_: C, 62.46; H, 6.05; N, 8.40%. Found: C, 62.01; H, 6.18; N, 8.56%.

3-phenyl-5-((4′-(2′’-oxopyrrolidin-1′’-yl)-3′,4′-dihydroquinolin-1′(2′H)-yl)methyl)isoxazole (6a): 55% yield; Orange oil; IR (ATR): 3465, 2931, 1670, 1497, 1438, 1167 cm^−1^. ^1^H NMR (400 MHz, CDCl_3_) δ (ppm): 7.76–7.72 (2H, m), 7.43–7.40 (3H, m), 7.10 (1H, tdd, *J* = 7.8, 1.6, 0.7 Hz), 6.89 (1H, td, *J* = 7.4, 1.1 Hz), 6.70 (1H, dd, *J* = 7.4, 1.0 Hz), 6.66 (1H, d, *J* = 8.6 Hz), 6.37 (1H, s), 5.41 (1H, dd, *J* = 9.2, 5.4 Hz), 4.62 (1H, d, J = 17.5 Hz), 4.56 (1H, d, J = 17.5 Hz), 3.60–3.39 (2H, m), 3.27–3.10 (2H, m), 2.51–2.47 (2H, m), 2.23–2.05 (2H, m), 2.05–1.95 (2H, m). ^13^C NMR (100 MHz, CDCl_3_) δ (ppm): 175.72, 170.00, 162.51, 144.97, 130.15, 128.97 (2C), 128.83, 128.80, 127.91, 126.87 (2C), 120.11, 117.11, 111.96, 100.38, 48.20, 47.97, 47.71, 43.83, 31.5, 26.70, 18.35. ESI-MS (m/z): 289.1 [M-C_4_H_8_NO]^+^, 374.1 [M + H]^+^, 396.1 [M + Na]^+^, 769.1 [2M + Na]^+^. Anal. Calcd. for C_23_H_23_N_3_O_2_: C, 73.97; H, 6.21; N, 11.25%. Found: C, 73.49; H, 6.34; N, 11.25%. The NMR and ESI-MS data match the previously reported data [21]. 

3-(4-methoxyphenyl)-5-((4′-(2′’-oxopyrrolidin-1′’-yl)-3′,4′-dihydroquinolin-1′(2′H)-yl)methyl)isoxazole (6b): 65% yield; Red oil; IR (ATR): 2961, 1672, 1422, 1250, 1017 cm^−1^. ^1^H NMR (400 MHz, CDCl_3_) δ (ppm): 7.67 (2H, d, *J* = 8.9 Hz), 7.09 (2H, td, *J* = 7.4, 1.2 Hz), 6.92 (2H, d, *J* = 8.9 Hz), 6.88 (1H, td, *J* = 7.7, 1.4 Hz), 6.68 (1H, dd, *J* = 7.4, 0.9 Hz), 6.64 (1H, d, *J* = 8.5 Hz), 6.32 (1H, s), 5.40 (1H, dd, *J* = 9.4, 5.6 Hz), 4.59 (1H, d, *J* = 17.7 Hz), 4.53 (1H, d, *J* = 17.7 Hz), 3.81 (3H, s), 3.59–3.37 (2H, m), 3.27–3.09 (2H, m), 2.48 (2H, td, J = 7.7, 1.0 Hz), 2.21–2.02 (2H, m), 2.08–1.91 (2H, m). ^13^C NMR (100 MHz, CDCl_3_) δ (ppm): 175.70, 169.68, 162.02, 161.01, 144.97, 128.67, 128.24 (2C), 127.82, 121.27, 120.04, 117.60, 114.18 (2C), 111.93, 100.10, 55.38, 48.14, 47.94, 47.64, 43.79, 31.47, 26.65, 18.31. ESI-MS (m/z): 426.1 [M + Na]^+^, 829.1 [2M + Na]^+^. Anal. Calcd. for C_24_H_25_N_3_O_3_: C, 71.44; H, 6.25; N, 10.41%. Found: C, 72.01; H, 6.12; N, 10.18%.

3-(3,4,5-trimethoxyphenyl)-5-((4′-(2′’-oxopyrrolidin-1′’-yl)-3′,4′-dihydroquinolin-1′(2′H)-yl)methyl)isoxazole (6c): 75% yield; Brown oil; IR (ATR): 3437, 2939, 1668, 1421, 1236, 1124 cm^−1^. ^1^H NMR (400 MHz, CDCl_3_) δ (ppm): 7.31–7.27 (1H, m), 7.06–6.92 (2H, m), 6.85–6.81 (1H, m), 6.64 (1H, d, *J* = 10.1 Hz), 6.56 (1H, d, *J* = 7.1 Hz), 6.32 (1H, s), 5.47–5.34 (1H, m), 4.64 (1H, d, *J* = 16.7 Hz), 4.51 (1H, d, *J* = 16.7 Hz), 3.91 (6H, s), 3.87 (3H, s), 3.70–3.36 (2H, m), 3.35–3.12 (2H, m), 2.56–2.44 (2H, m), 2.25–1.89 (4H, m). ^13^C NMR (100 MHz, CDCl_3_) δ (ppm): 174.98, 169.49, 162.41, 153.65 (2C), 143.46, 139.65, 128.61, 127.12, 124.11, 122.55, 121.96, 113.17, 104.05 (2C), 100.49, 61.02, 56.44 (2C), 49.98, 48.41, 47.82, 43.53, 31.24, 26.38, 18.33. ESI-EM (m/z): 486.0 [M + Na]^+^, 520.1 [M + Cl + Na]^+^. Anal. Calcd for C_26_H_29_N_3_O_5_: C, 67.37; H, 6.31; N, 9.07%. Found: C, 67.78; H, 6.17; N, 8.81%. The NMR and ESI-MS data match the previously reported data [21].

5-((6′-methyl-4′-(2′’-oxopyrrolidin-1′’-yl)-3′,4′-dihydroquinolin-1′(2′H)-yl)methyl)-3-phenylisoxazole (6d): 72% yield; Red oil; IR (ATR): 2952, 1667, 1421, 1285, 1093 cm^−1^. ^1^H NMR (400 MHz, CDCl_3_) δ (ppm): 7.76–7.72 (2H, m), 7.43–7.40 (3H, m), 6.91 (1H, dd, *J* = 8.4, 2.1 Hz), 6.71 (1H, d, *J* = 1.8 Hz), 6.58 (1H, d, *J* = 8.4 Hz), 6.36 (1H, s), 5.38 (1H, dd, *J* = 9.0, 5.4 Hz), 4.59 (1H, d, *J* = 17.4 Hz), 4.54 (1H, d, *J* = 17.4 Hz), 3.56–3.36 (2H, m), 3.27–3.10 (2H, m), 2.52–2.48 (2H, m), 2.19 (3H, s), 2.24–2.15 (2H, m), 2.10–1.95 (2H, m). ^13^C NMR (100 MHz, CDCl_3_) δ (ppm): 175.66, 170.18, 162.52, 142.82, 130.15, 129.37, 128.98 (2C), 128.90, 128.51, 127.04, 126.90 (2C), 120.18, 112.23, 100.41, 48.23, 47.83, 47.87, 43.93, 31.56, 26.98, 20.46, 18.41. ESI-MS (m/z): 410.1 [M + Na]^+^, 797.2 [2M + Na]^+^. Anal. Calcd for C_24_H_25_N_3_O_2_: C, 74.39; H, 6.50; N, 10.84%. Found: C, 74.78; H, 6.37; N, 10.61%. The NMR and ESI-MS data match the previously reported data [21].

3-(4-methoxyphenyl)-5-((6′-methyl-4′-(2′’-oxopyrrolidin-1′’-yl)-3′,4′-dihydroquinolin-1′(2′H)-yl)methyl)isoxazole (6e): 64% yield; Red oil; IR (ATR): 2957, 1664, 1425, 1251, 1176 cm^−1^. ^1^H NMR (400 MHz, CDCl_3_) δ (ppm): 7.72 (2H, dd, *J* = 6.1, 2.1 Hz), 7.10 (1H, d, *J* = 1.9 Hz), 6.94 (2H, dd, *J* = 6.1, 1.9 Hz), 6.89–6.85 (1H, m), 6.71 (1H, d, *J* = 1.8 Hz), 6.59 (1H, s), 6.32 (1H, s), 5.36 (1H, dd, *J* = 9.2, 7.2 Hz), 4.38 (1H, d, *J* = 16.5 Hz), 4.31 (1H, d, *J* = 16.5 Hz), 3.84 (3H, s), 3.39–3.22 (2H, m), 3.20–2.96 (2H, m), 2.54–2.43 (2H, m), 2.21 (2H, s), 2.12–1.81 (4H, m). ^13^C NMR (100 MHz, CDCl_3_) δ (ppm): 175.84, 171.14, 162.22, 161.99, 142.23, 133.44, 130.68, 129.08, 128.23 (2C), 127.71, 126.88, 121.46, 114.31 (2C), 100.75, 55.39, 48.69, 47.70, 46.96, 43.03, 31.21, 26.33, 20.60, 18.16. ESI-MS (m/z): 474.1 [M + Cl + Na]+, 925.1 [2M + 2Cl + Na]+. Anal. Calcd for C_25_H_27_N_3_O_3_: C, 71.92; H, 6.52; N, 10.06%. Found: C, 73.72; H, 6.68; N, 9.86%. The NMR and ESI-MS data match the previously reported data [20].

5-((8′-chloro-6′-methyl-4′-(2′’-oxopyrrolidin-1′’-yl)-3′,4′-dihydroquinolin-1′(2′H)-yl)methyl)-3-(3,4-dimethoxyphenyl)isoxazole (6f): 58% yield; Beige solid; IR (ATR): 2966, 1656, 1429, 1255, 1143, 1016 cm^−1^. ^1^H NMR (400 MHz, CDCl_3_) δ (ppm): 7.40 (1H, d, J = 8.3 Hz), 7.30 (1H, dd, J = 8.3, 2.0 Hz), 7.11 (1H, d, J = 2.1 Hz), 6.91 (1H, d, J = 8.3 Hz), 6.73 (1H, t, J = 1.0 Hz), 6.63 (1H, s), 5.38 (1H, dd, J = 8.8, 7.1 Hz), 4.38 (1H, d, J = 16.3 Hz), 4.33 (1H, d, J = 16.3 Hz), 3.94 (3H, s), 3.91 (3H, s), 3.31–3.21 (2H, m), 3.24–3.00 (2H, m), 2.51–2.45 (2H, m), 2.23 (3H, s), 2.06–1.95 (2H, m), 1.94–1.84 (2H, m). ^13^C NMR (100 MHz, CDCl_3_) δ (ppm): 175.75, 171.34, 162.42, 150.62, 149.30, 142.30, 133.53, 130.73, 129.18, 127.78, 126.96, 121.75, 120.03, 111.05, 109.24, 100.78, 56.09, 56.02, 49.82, 47.70, 46.98, 43.05, 31.28, 21.41, 20.67, 18.24. ESI-MS (m/z): 482.0 [M + H]^+^, 504.1 [M + Na]^+^, 985.1 [2M + Na]^+^. Anal. Calcd for C_26_H_28_ClN_3_O_4_: C, 64.79; H, 5.86; N, 8.72%. Found: C, 64.58; H, 5.75; N, 8.61%

3-(3,4,5-trimethoxyphenyl)-5-((6′-methyl-4′-(2′’-oxopyrrolidin-1′’-yl)-3′,4′-dihydroquinolin-1′(2′H)-yl)methyl)isoxazole (6g): 63% yield; Red oil; IR (ATR): 2935, 1668, 1421, 1124, 1001 cm^−1^. ^1^H NMR (400 MHz, CDCl_3_) δ (ppm): 6.96 (2H, s), 6.91 (1H, dd, *J* = 8.4, 2.0 Hz), 6.7 (1H, d, *J* = 2.0 Hz), 6.56 (1H, d, *J* = 8.4 Hz), 6.32 (1H, s), 5.40 (1H, dd, *J* = 9.2, 5.6 Hz), 4.62 (1H, d, *J* = 17.6 Hz), 4.51 (1H, d, *J* = 17.6 Hz), 3.90 (6H, s), 3.86 (3H, s), 3.58–3.37 (2H, m), 3.28–3.12 (2H, m), 2.55–2.47 (2H, m), 2.19 (3H, s), 2.23–2.06 (2H,m), 2.05–1.95 (2H, m). ^13^C NMR (100 MHz, CDCl_3_) δ (ppm): 175.75, 170.33, 162.43, 153.65 (2C), 142.81, 139.64, 133.64, 129.39, 128.34, 124.57, 120.19, 112.16, 104.08 (2C), 100.29, 61.06, 56.42 (2C), 48.37, 48.00, 47.99, 43.83, 31.55, 26.92, 20.48, 18.42. ESI-MS (m/z): 391.1 [M-C_4_H_7_NO]^+^, 478.3 [M + H]^+^, 500.2 [M + Na]^+^, 539.1 [M+C_2_H_8_CONa]^+^. Anal. Calcd for C_27_H_31_N_3_O_5_: C, 67.91; H, 6.54; N, 8.80%. Found: C, 68.33; H, 6.69; N, 9.04%. The NMR and ESI-MS data match the previously reported data [21].

5-((6′-methoxy-4′-(2′’-oxopyrrolidin-1′’-yl)-3′,4′-dihydroquinolin-1′(2′H)-yl)methyl)-3-phenylisoxazole (6h): 42% yield; Brown oil; IR (ATR): 2949, 1661, 1502, 1421, 1285, 1039 cm^−1^. ^1^H NMR (400 MHz, CDCl_3_) δ (ppm): 7.74–7.72 (2H, m), 7.45–7.39 (3H, m), 6.69 (1H, dd, *J* = 2.9, 0.7 Hz), 6.61 (1H, d, *J* = 9.0 Hz), 6.49 (1H, dd, *J* = 2.9, 0.7 Hz), 6.35 (1H, s), 5.39 (1H, dd, *J* = 9.3, 5.6 Hz), 4.57 (1H, d, *J* = 17.4 Hz), 4.49 (1H, d, J = 17.4 Hz), 3.68 (3H, s), 3.51–3.32 (2H, m), 3.26–3.10 (2H, m), 2.50–2.45 (2H, m), 2.19–2.06 (2H, m), 2.06–1.93 (2H, m). ^13^C NMR (100 MHz, CDCl_3_) δ (ppm): 175.69, 170.16, 162.46, 152.08, 139.43, 130.12, 128.95 (2C), 128.84, 126.85 (2C), 121.77, 114.12, 113.68, 113.48, 100.45, 55.78, 48.70, 48.35, 48.16, 43.68, 31.45, 26.85, 18.36. ESI-MS (m/z): 426.1 [M + Na]^+^, 829.1 [2M + Na]^+^. Anal. Calcd. for C_24_H_25_N_3_O_3_ (403.48 g/mol). Anal. Calcd for C_24_H_25_N_3_O_3_: C, 71.44; H, 6.25; N, 10.41%. Found: C, 71.08; H, 6.39; N, 10.13%. The NMR and ESI-MS data match the previously reported data [21].

3-(4-methoxyphenyl)-5-((6′-methoxy-4′-(2′’-oxopyrrolidin-1′’-yl)-3′,4′-dihydroquinolin-1′(2′H)-yl)methyl)isoxazole (6i): 74% yield; Red oil; IR (ATR): 2937, 1662, 1427, 1249, 1174 cm^−1^. ^1^H NMR (400 MHz, CDCl_3_) δ (ppm): 7.67 (2H, dd, *J* = 6.8, 2.1 Hz), 6.92 (2H, dd, *J* = 6.8, 1.9 Hz), 6.70 (1H, dd, *J* = 8.9, 2.7 Hz) 6.61 (1H, d, *J* = 8.9 Hz), 6.49 (1H, d, *J* = 0.9 Hz), 6.29 (1H, s), 5.36 (1H, dd, *J* = 8.9, 6.5 Hz), 4.56 (1H, d, *J* = 17.6 Hz), 4.48 (1H, d, *J* = 17.6 Hz), 3.82 (3H, s), 3.69 (3H, s), 3.51–3.32 (2H, m), 3.18–3.00 (2H, m), 2.50–2.45 (2H, m), 2.20–2.04 (2H, m), 2.09–1.91 (2H, m). ^13^C NMR (100 MHz, CDCl_3_) δ (ppm): 175.72, 169.80, 161.65, 152.08, 138.32, 130.40, 128.98, 128.20 (2C), 121.23, 114.32 (2C), 114.14, 113.51, 111.72, 100.22, 55.77, 55.42, 48.17, 47.99, 47.00, 43.98, 31.23, 26.87, 18.38. ESI-MS (m/z): 456.1 [M + Na]^+^, 889.2 [2M + Na]^+^. Anal. Calcd for C_25_H_27_N_3_O_4_: C, 69.27; H, 6.28; N, 9.69%. Found: C, 68.87; H, 6.15; N, 9.44%.

5-((8′-chloro-6′-methoxy-4′-(2′’-oxopyrrolidin-1′’-yl)-3′,4′-dihydroquinolin-1′(2′H)-yl)methyl)-3-(3,4-dimethoxyphenyl)isoxazole (6j): 86% yield; Beige solid; IR (ATR): 2943, 2839, 1670, 1429, 1261 cm^−1^. ^1^H NMR (400 MHz, CDCl_3_) δ (ppm): 7.40 (1H, d, J = 2.0 Hz), 7.29 (1H, dd, J = 8.5, 2.0 Hz), 6.91 (1H, d, J = 8.5 Hz), 6.90 (1H, dd, J = 2.8, 0.8 Hz), 6.62 (1H, s), 6.50 (1H, dd, J = 2.9, 0.9 Hz), 5.39 (1H, t, J = 8.2 Hz), 4.34 (1H, d, J = 16.3 Hz), 4.28 (1H, d, J = 16.3 Hz), 3.93 (3H, s), 3.91 (3H, s), 3.72 (3H, s), 3.40–3.19 (2H, m), 3.27–3.01 (2H, m), 2.49–2.45 (2H, m), 2.15–1.96 (2H, m), 1.95–1.83 (2H, m). ^13^C NMR (100 MHz, CDCl_3_) δ (ppm): 175.71, 171.32, 162.42, 155.47, 150.63, 149.31, 138.32, 130.42, 128.98, 121.74, 120.03, 115.91, 111.73, 111.06, 109.25, 100.78, 56.09, 56.02, 55.77 49.86, 47.96, 46.95, 42.97, 31.23, 21.58, 18.28. ESI-MS (m/z): 486.2 [[M-Cl]+Na]^+^, 520.1 [M + Na]^+^, 1017.2 [2M + Na]^+^. Anal. Calcd for C_26_H_28_ClN_3_O_5_: C, 62.71; H, 5.67; N, 8.44%. Found: C, 62.54; H, 5.57; N, 8.67%.

3-(3,4,5-trimethoxyphenyl)-5-((6′-methoxy-4′-(2′’-oxopyrrolidin-1′’-yl)-3′,4′-dihydroquinolin-1′(2′H)-yl)methyl)isoxazole (6k): 74% yield; Red oil; IR (ATR): 2937, 1664, 1421, 1238, 1124 cm^−1^. ^1^H NMR (400 MHz, CDCl_3_) δ (ppm): 6.97 (2H, s), 6.71 (1H, dd, *J* = 9.0, 2.9 Hz), 6.61 (1H, d, *J* = 9.0 Hz), 6.50 (1H, dd, *J* = 2.9, 0.8 Hz), 6.31 (1H, s), 5.43 (1H, dd, *J* = 9.6, 5.7 Hz), 4.62 (1H, d, *J* = 17.4 Hz), 4.48 (1H, d, *J* = 17.4 Hz), 3.90 (6H, s), 3.87 (3H, s), 3.70 (3H, s), 3.56–3.35 (2H, m), 3.28–3.14 (2H, m), 2.53–2.48 (2H, m), 2.21–2.08 (2H,m), 2.07–1.97 (2H, m). ^13^C NMR (100 MHz, CDCl_3_) δ (ppm): 175.77, 170.34, 162.44, 153.68 (2C), 152.18, 139.67, 139.43, 124.38, 121.88, 114.12, 113.63, 113.44, 104.10 (2C), 100.38, 61.09, 56.44 (2C), 55.86, 48.57, 48.35, 48.19, 43.64, 31.51, 26.86, 18.44. ESI-MS (m/z): 516.1 [M + Na]^+^, 1009.2 [2M + Na]^+^. Anal. Calcd for C_27_H_31_N_3_O_6_: C, 65.71; H, 6.33; N, 8.51%. Found: C, 65.49; H, 6.21; N, 8.30%. The NMR and ESI-MS data match the previously reported data [21].

5-((6′-chloro-4′-(2′’-oxopyrrolidin-1′’-yl)-3′,4′-dihydroquinolin-1′(2′H)-yl)methyl)-3-phenylisoxazole (6l): 55% yield; Red oil; IR (ATR): 2930, 1657, 1496, 1127 cm^−1^. ^1^H NMR (400 MHz, CDCl_3_) δ (ppm): 7.77–7.72 (2H, m), 7.44–7.41 (3H, m), 7.04 (1H, dd, *J* = 8.9, 2.5 Hz), 6.83 (1H, dd, *J* = 2.5, 1.0 Hz), 6.57 (1H, d, *J* = 8.9 Hz), 6.37 (1H, s), 5.37 (1H, dd, *J* = 9.6, 5.9 Hz), 4.60 (1H, d, *J* = 17.5 Hz), 4.52 (1H, d, *J* = 17.5 Hz), 3.60–3.39 (2H, m), 3.28–3.11 (2H, m), 2.55–2.44 (2H, m), 2.21–2.03 (2H, m), 2.09–1.98 (2H, m). ^13^C NMR (100 MHz, CDCl_3_) δ (ppm): 175.77, 169.43, 162.55, 143.55, 130.25, 129.01 (2C), 128.69, 128.60, 127.28, 126.87 (2C), 122.56, 121.90, 113.23, 100.47, 48.24, 47.75, 47.69, 43.54, 31.15, 26.41, 18.34. ESI-MS (m/z): 430.0 [M + Na]^+^, 837.1 [2M + Na]^+^. Anal. Calcd for C_23_H_22_ClN_3_O_2_: C, 67.73; H, 5.44; N, 10.30%. Found: C, 67.36; H, 5.58; N, 10.09%. The NMR and ESI-MS data match the previously reported data [21].

5-((6′-chloro-4′-(2′’-oxopyrrolidin-1′’-yl)-3′,4′-dihydroquinolin-1′(2′H)-yl)methyl)-3-(4-methoxyphenyl)isoxazole (6m): 51% yield; Orange oil; IR (ATR): 2976, 1678, 1431, 1253, 1178 cm^−1^. ^1^H NMR (400 MHz, CDCl_3_) δ (ppm): 7.67 (2H, dd, *J* = 6.1, 2.2 Hz), 7.03 (1H, ddd, *J* = 8.9, 2.6, 0.8 Hz), 6.93 (2H, dd, *J* = 6.1, 2.2 Hz), 6.83 (1H, dd, *J* = 2.6, 1.0 Hz), 6.57 (1H, d, *J* = 8.9 Hz), 6.31 (1H, s), 5.37 (1H, dd, *J* = 9.6, 5.4 Hz), 4.58 (1H, d, *J* = 17.6 Hz), 4.50 (1H, d, *J* = 17.6 Hz), 3.82 (3H, s), 3.60–3.39 (2H, m), 3.28–3.12 (2H, m), 2.56–2.46 (2H, m), 2.19–2.06 (2H, m), 2.09–2.01 (2H, m). ^13^C NMR (100 MHz, CDCl_3_) δ (ppm): 175.94, 169.14, 162.16, 161.14, 143.60, 128.64, 128.30 (2C), 127.30, 122.54, 121.82, 121.19, 114.37 (2C), 113.27, 100.25, 55.44, 48.21, 47.83, 47.70, 43.68, 31.36, 26.43, 18.16. ESI-MS (m/z): 460.1 [M + Na]^+^, 897.1 [2M + Na]^+^. Anal. Calcd for C_24_H_24_ClN_3_O_3_: C, 65.83; H, 5.52; N, 9.60%. Found: C, 67.47; H, 5.66; N, 9.48%. The NMR and ESI-MS data match the previously reported data [20].

5-((6′-chloro-4′-(2′’-oxopyrrolidin-1′’-yl)-3′,4′-dihydroquinolin-1′(2′H)-yl)methyl)-3-(3,4-dimethoxyphenyl)isoxazole (6n): 68% yield; Orange oil; IR (ATR): 2935, 1665, 1420, 1257, 1022 cm^−1^. ^1^H NMR (400 MHz, CDCl_3_) δ (ppm): 7.34 (1H, d, *J* = 2.0 Hz), 7.22 (1H, dd, *J* = 8.3, 2.2 Hz), 7.02 (1H, dd, *J* = 8.8, 2.5 Hz), 6.87 (1H, d, *J* = 8.3 Hz), 6.82 (1H, dd, *J* = 2.5, 1.0 Hz), 6.56 (1H, d, *J* = 8.8 Hz), 6.31 (1H, s), 5.36 (1H, dd, *J* = 9.7, 5.3 Hz), 4.59 (1H, d, *J* = 17.6 Hz), 4.49 (1H, d, *J* = 17.6 Hz), 3.91 (3H, s), 3.89 (3H, s), 3.60–3.36 (2H, m), 3.27–3.11 (2H, m), 2.55–2.44 (2H, m), 2.21–2.10 (2H, m), 2.09–1.98 (2H, m). ^13^C NMR (100 MHz, CDCl_3_) δ (ppm): 175.76, 169.20, 162.24, 150.68, 149.27, 143.54, 128.57, 127.19, 122.50, 121.87, 121.36, 120.05, 113.20, 110.97, 109.16, 100.22, 56.07, 55.98, 48.25, 47.75, 47.68, 43.56, 31.32, 26.36, 18.34. ESI-MS (m/z): 456.1 [[M-Cl]+H]^+^, 490.1 [M + Na]^+^, 597.1 [2M + Na]^+^. Anal. Calcd for C_25_H_26_ClN_3_O_4_: C, 64.17; H, 5.60; N, 8.98%. Found: C, 63.91; H, 5.49; N, 8.81%.

5-((6′,8′-dichloro-4′-(2′’-oxopyrrolidin-1′’-yl)-3′,4′-dihydroquinolin-1′(2′H)-yl)methyl)-3-(3,4-dimethoxyphenyl)isoxazole (6o): 88% yield; Orange solid; IR (ATR): 2954, 1662, 1427, 1249 cm^−1^. ^1^H NMR (400 MHz, CDCl_3_) δ (ppm): 7.40 (1H, d, *J* = 2.0 Hz), 7.30 (1H, d, *J* = 0.8 Hz), 7.29 (1H, dd, *J* = 8.3, 2.0 Hz), 6.92 (1H, d, *J* = 0.8 Hz), 6.91 (1H, d, *J* = 8.4 Hz), 6.61 (1H, s), 5.39 (1H, t, *J* = 8.3 Hz), 4.43 (1H, d, *J* = 16.4 Hz), 4.36 (1H, d, *J* = 16.4 Hz), 3.94 (3H, s), 3.92 (3H, s), 3.32–3.24 (2H, m), 3.25–3.01 (2H, m), 2.55–2.42 (2H, m), 2.06–1.97 (2H, m), 1.96–1.85 (2H, m). ^13^C NMR (100 MHz, CDCl_3_) δ (ppm): 175.83, 170.79, 162.46, 150.73, 149.37, 143.62, 130.42, 129.95, 128.61, 128.25, 126.29, 121.63, 120.08, 111.08, 109.24, 100.93, 56.13, 56.06, 49.44, 47.80, 47.27, 42.96, 31.13, 21.41, 18.28. ESI-MS (m/z): 490.1 [[M-Cl] + Na]^+^, 524.0 [M + Na]^+^, 1027.0 [2M + Na]^+^. Anal. Calcd for C_25_H_25_Cl_2_N_3_O_4_: C, 59.77; H, 5.02; N, 8.36%. Found: C, 59.52; H, 4.91; N, 8.46%.

5-((6′-chloro-4′-(2′’-oxopyrrolidin-1′’-yl)-3′,4′-dihydroquinolin-1′(2′H)-yl)methyl)-3-(3,4,5-trimethoxyphenyl)isoxazole (6p): 85% yield; Brown oil; IR (ATR): 2936, 1667, 1582, 1420, 1122 cm^−1^. ^1^H NMR (400 MHz, CDCl_3_) δ (ppm): 7.05 (1H, dd, *J* = 8.8, 2.6 Hz), 6.96 (2H, s), 6.84 (1H, dd, *J* = 2.5, 0.9 Hz), 6.57 (1H, d, *J* = 8.8 Hz), 6.31 (1H, s), 5.40 (1H, dd, *J* = 9.9, 5.4 Hz), 4.64 (1H, d, *J* = 17.6 Hz), 4.51 (1H, d, *J* = 17.6 Hz), 3.90 (6H, s), 3.87 (3H, s), 3.64–3.40 (2H, m), 3.30–3.14 (2H, m), 2.59–2.44 (2H, m), 2.23–2.05 (2H, m), 2.16–1.97 (2H, m). ^13^C NMR (100 MHz, CDCl_3_) δ (ppm): 175.82, 169.55, 160.50, 153.72 (2C), 143.55, 139.80, 128.67, 127.23, 124.20, 122.69, 122.02, 113.22, 104.14 (2C), 100.39, 61.08, 56.45 (2C), 48.44, 47.86, 47.84, 43.54, 31.38, 26.40, 18.40. ESI-MS (m/z): 520.1 [M + Na]^+^, 1017.2 [2M + Na]+, 1513.0 [3M + Na]^+^. Anal. Calcd for C_26_H_28_ClN_3_O_5_: C, 62.71; H, 5.67; N, 8.44%. Found: C, 63.12; H, 5.53; N, 8.68%. The NMR and ESI-MS data match the previously reported data [21].

5-((6′-ethyl-4′-(2′’-oxopyrrolidin-1′’-yl)-3′,4′-dihydroquinolin-1′(2′H)-yl)methyl)-3-phenylisoxazole (6q): 40% yield; Brown oil; IR (ATR): 2863, 2363, 1681, 1510, 1335, 1169 cm^−1^. ^1^H NMR (400 MHz, CDCl_3_) δ (ppm): 7.75 (2H, d, *J* = 4.8 Hz), 7.42 (3H, br), 6.95 (1H, d, *J* = 8.3 Hz), 6.74 (1H, s), 6.61 (1H, d, *J* = 8.3 Hz), 6.38 (1H, s), 5.44–5.38 (1H, m), 4.63–4.52 (2H, m), 3.53 (1H, t, *J* = 10,1 Hz), 3.44–3.37 (1H, m), 3.24 (1H, dd, *J* = 16.0, 8.0 Hz), 3.13 (1H, dd, *J* = 14.6, 8.0 Hz), 2.52 (4H, bs), 2.24–2.06 (2H, m), 2.03–1.93 (2H, m), 1.15 (3H, t, *J* = 7.5 Hz). ^13^C NMR (100 MHz, CDCl_3_) δ (ppm): 175.91, 170.61, 162.88, 143.43, 134.01, 130.47, 129.32 (2C), 128.99, 128.49, 127.79, 127.26 (2C), 120.56, 112.57, 100.75, 48.59, 48.37, 48.29, 44.33, 31.95, 28.24, 27.39, 18.87, 16.31. HR-ESI-MS (m/z): 402.2156 [M + H]^+^, 420.2031 [M + Na]^+^, 440.1745 [M + K]^+^. Anal. Calcd. for C_25_H_27_N_3_O_2_ (401.2103 g/mol).

5-((6′-ethyl-4′-(2′’-oxopyrrolidin-1′’-yl)-3′,4′-dihydroquinolin-1′(2′H)-yl)methyl)-3-(4-methoxyphenyl)isoxazole (6r): 60% yield; Orange oil; IR (ATR): 2961, 2358, 1963, 1679, 1507, 1436, 1255, 1027 cm^−1^. ^1^H NMR (400 MHz, CDCl_3_) δ (ppm): 7.79 (2H, d, *J* = 8.1 Hz), 7.36 (1H, s), 7.04 (2H, d, *J* = 8.2 Hz), 6.84 (1H, s), 6.71 (1H, d, *J* = 8.2 Hz), 6.42 (1H, s), 5.54–5.48 (1H, m), 4.66 (2H, s), 3.94 (3H, s), 3.63 (1H, t, *J* = 10.1 Hz), 3.53–3.46 (1H, m), 3.34 (1H, dd, *J* = 16.1, 8.0 Hz), 3.23 (1H, dd, *J* = 13.9, 8.7 Hz), 2.64–2.54 (4H, m), 2.33–2.23 (2H, m), 2.13–2.03 (2H, m), 1.25 (3H, t, *J* = 7.5 Hz). ^13^C NMR (100 MHz, CDCl_3_) δ (ppm): 175.98, 170.32, 162.50, 161.47, 143.48, 133.98, 128.68 (2C), 128.51, 127.79, 121.85, 120.52, 114.73 (2C), 112.60, 100.52, 55.81, 48.57, 48.42, 48.30, 44.39, 31.97, 28.25, 27.40, 18.88, 16.32. HR-ESI-MS (m/z): 432.2271 [M + H]^+^, 346.0085 [M-C_4_H_7_NO]^+^. Anal. Calcd. for C_26_H_29_N_3_O_3_ (431.2209 g/mol).

5-((8′-chloro-6′-ethyl-4′-(2′’-oxopyrrolidin-1′’-yl)-3′,4′-dihydroquinolin-1′(2′H)-yl)methyl)-3-(3,4-dimethoxyphenyl)isoxazole (6s): 73% yield; Beige solid; IR (ATR): 2966, 2360, 1679, 1470, 1267, 1019 cm^−1^. ^1^H NMR (400 MHz, CDCl_3_) δ (ppm): 7.41 (1H, s), 7.30 (1H, d, *J* = 8.2 Hz), 7.15 (1H, s), 6.92 (1H, d, *J* = 8.2 Hz), 6.77 (1H, s), 6.62 (1H, s), 5.42 (1H, m), 4.45–4.33 (2H, m), 3.95 (3H, s), 3.92 (3H, s), 3.34–3.23 (2H, m), 3.22–2.99 (2H, m), 2.58–2.51 (2H, m), 2.50–2.45 (2H, m), 2.05–1.95 (2H, m), 1.95–1.88 (2H, m), 1.18 (3H, t, *J* = 7.5 Hz). ^13^C NMR (100 MHz, CDCl_3_) δ (ppm): 176.00, 171.73, 162.79, 151.07, 149.77, 142.88, 140.25, 129.93, 129.64, 128.21, 126.25, 122.23, 120.41, 111.53, 109.78, 101.14, 56.49, 56.41, 50.21, 48.21, 47.42, 43.47, 31.68, 28.39, 21.94, 18.74, 15.94. HR-ESI-MS (m/z): 496.2019 [M + H]^+^, 518.1739 [M + Na]^+^, 534.1592 [M + K]^+^. Anal. Calcd. for C_27_H_30_ClN_3_O_4_ (495.1925 g/mol).

5-((6′-ethyl-4′-(2′’-oxopyrrolidin-1′’-yl)-3′,4′-dihydroquinolin-1′(2′H)-yl)methyl)-3-(3,4,5- trimethoxyphenyl)isoxazole (6t): 72% yield; Orange oil; IR (ATR): 2961, 2929, 1679, 1581, 1230, 1127 cm^−1^. ^1^H NMR (400 MHz, CDCl_3_) δ (ppm): 7.13 (2H, br), 7.10 (1H, d, *J* = 7.7 Hz), 6.88 (1H, s), 6.76 (1H, d, *J* = 8.4 Hz), 6.52 (1H, s), 5.59–5.53 (1H, m), 4.76 (1H, d, *J* = 17.4 Hz), 4.68 (1H, d, *J* = 17.5 Hz), 4.05 (6H, s), 4.02 (3H, s), 3.75–3.52 (2H, m), 3.45–3.26 (2H, m), 2.70–2.62 (4H, m), 2.50–2.32 (2H, m), 2.28–2.21 (2H, m), 1.30 (3H, t, *J* = 7.5 Hz). ^13^C NMR (100 MHz, CDCl_3_) δ (ppm): 176.16, 170.61, 162.61, 153.87 (2C), 143.27, 139.97, 133.85, 128.38, 127.45, 124.58, 120.28, 112.43, 104.45 (2C), 100.57, 61.22, 56.63 (2C), 48.49, 48.38, 48.17, 44.22, 31.76, 28.08, 27.14, 18.68, 16.17. HR-ESI-MS (m/z): 492.2448 [M + H]^+^. Anal. Calcd. for C_28_H_33_N_3_O_5_ (491.2420 g/mol).

5-((6′-fluor-4′-(2′’-oxopyrrolidin-1′’-yl)-3′,4′-dihydroquinolin-1′(2′H)-yl)methyl)-3-phenylisoxazole (6u): 89% yield; Orange oil; IR (ATR): 2941, 2870, 1665, 1499, 1281, 1155 cm^−1^. ^1^H NMR (400 MHz, CDCl_3_) δ (ppm): 7.75 (2H, d, *J* = 2.7 Hz), 7.43 (3H, br), 6.83 (1H, t, *J* = 8.4 Hz), 6.67–6.58 (2H, m), 6.37 (1H, s), 5.45–5.39 (1H, m), 4.61 (1H, d, *J* = 17.4 Hz), 4.53 (1H, d, *J* = 17.4 Hz), 3.56 (1H, t, *J* = 10.7 Hz), 3.44–3.37 (1H, m), 3.27 (1H, dd, *J* = 16.2, 8.1 Hz), 3.15 (1H, dd, *J* = 14.6, 7.8 Hz), 2.55–2.46 (2H, m), 2.23–2.07 (2H, m), 2.06–2.00 (2H, m). ^13^C NMR (100 MHz, CDCl_3_) δ (ppm): 175.56, 169.65, 162.46, 156.94, 154.59, 141.44, 130.11, 128.91 (2C), 126.81 (2C), 122.02, 115.64 (JC-F = 22.2), 114. 52 (JC-F = 22.2), 113.10 (JC-F = 7.2), 100.38, 48.38, 48.07, 47.86, 43.38, 31.26, 26.49, 18.27. HR-ESI-MS (m/z): 392.1917 [M + H]^+^, 414.1691 [M + Na]^+^, 430.1480 [M + K]^+^. Anal. Calcd. for C_23_H_22_FN_3_O_2_ (391.1696 g/mol).

5-((6′-fluoro-4′-(2′’-oxopyrrolidin-1′’-yl)-3′,4′-dihydroquinolin-1′(2′H)-yl)methyl)-3-(4-methoxyphenyl)isoxazole (6v): 77% yield; Orange oil; IR (ATR): 2954, 2836, 1669, 1608, 1429, 1252, 1027 cm^−1^. ^1^H NMR (400 MHz, CDCl_3_) δ (ppm): 7.69 (d, *J* = 7.9 Hz, 1H), 6.94 (d, *J* = 7.9 Hz, 1H), 6.82 (t, *J* = 8.4 Hz, 1H), 6.66–6.57 (m, 1H), 6.30 (s, 1H), 5.45–5.37 (m, 1H), 4.58 (d, *J* = 17.4 Hz, 1H), 4.51 (d, *J* = 17.3 Hz, 1H), 3.84 (s, 1H), 3.59–3.36 (m, 1H), 3.31–3.11 (m, 1H), 2.58–2.44 (m, 1H), 2.23–2.06 (m, 1H). ^13^C NMR (100 MHz, CDCl_3_) δ (ppm): 175.57, 169.36, 162.06, 161.07, 156.92, 141.47, 128.21 (2C), 122.00, 121.25, 115.18 (JC-F = 19.3), 114.30 (2C), 114.05 (JC-F = 22.6), 113.12 (JC-F = 7.3), 100.13, 55.36, 48.35, 48.05, 47.87, 43.39, 31.26, 26.48, 18.27. HR-ESI-MS (m/z): 422.1874 [M + H]^+^, 444.1710 [M + Na]^+^. Anal. Calcd. for C_24_H_24_FN_3_O_3_ (421.1802 g/mol).

5-((6′-fluoro-4′-(2′’-oxopyrrolidin-1′’-yl)-3′,4′-dihydroquinolin-1′(2′H)-yl)methyl)-3-(3,4-dimethoxyphenyl)isoxazole (6w): 75% yield; Orange oil; IR (ATR): 2934, 2834, 1679, 1502, 1264, 1024 cm^−1^. ^1^H NMR (400 MHz, CDCl_3_) δ (ppm): 7.40 (1H, s), 7.29 (1H, d, *J* = 5.8 Hz), 6.93 (1H, d, *J* = 8.2 Hz), 6.86 (1H, t, *J* = 8.4 Hz), 6.65 (2H, m), 6.35 (1H, s), 5.49–5.42 (1H, m), 4.63 (1H, d, *J* = 17.4 Hz), 4.54 (1H, d, *J* = 17.4 Hz), 3.97 (3H, s), 3.95 (3H, s), 3.59 (1H, t, *J* = 10.7 Hz), 3.47–3.40 (1H, m), 3.30 (1H, dd, *J* = 16.1, 8.1 Hz), 3.19 (1H, dd, *J* = 14.7, 7.6 Hz), 2.54 (2H, m), 2.27–2.10 (2H, m), 2.10–2.04 (2H, m). ^13^C NMR (100 MHz, CDCl_3_) δ (ppm): 176.05, 169.90, 162.63, 157.38, 155.02, 151.16, 149.77, 141.88, 121.91, 120.44, 115,62 (JC-F = 22.1), 114.46 (JC-F = 22.6), 113.55 (JC-F = 7.4), 111.48, 109.73, 100.61, 56.50, 56.40, 48.84, 48.52, 48.34, 43.81, 31.70, 26.90, 18.71. HR-ESI-MS (m/z): 452.1956 [M + H]^+^, 474.1831 [M + Na]^+^. Anal. Calcd. for C_25_H_26_FN_3_O_4_ (451.1907 g/mol).

5-((6′-fluoro-4′-(2′’-oxopyrrolidin-1′’-yl)-3′,4′-dihydroquinolin-1′(2′H)-yl)methyl)-3-(3,4,5- trimethoxyphenyl)isoxazole (6x): 87% yield; Orange oil; IR (ATR): 2935, 2357, 1661, 1504, 1254, 1126 cm^−1^. ^1^H NMR (400 MHz, CDCl_3_) δ (ppm): 6.97 (2H, s), 6.82 (1H, t, *J* = 8.4 Hz), 6.63 (1H, d, *J* = 9.0 Hz), 6.59 (1H, dd, *J* = 8.8, 4.2 Hz), 6.31 (1H, s), 5.43 (1H, m), 4.63 (1H, d, *J* = 17.5 Hz), 4.50 (1H, d, *J* = 17.4 Hz), 3.91 (6H, s), 3.88 (3H, s), 3.58 (1H, t, *J* = 10.8 Hz), 3.44–3.37 (1H, m), 3.28 (1H, dd, *J* = 16.1, 8.1 Hz), 3.17 (1H, dd, *J* = 14.7, 7.8 Hz), 2.54–2.47 (2H, m), 2.25–2.08 (2H, m), 2.06–2.00 (2H, m). ^13^C NMR (100 MHz, CDCl_3_) δ (ppm): 176.07, 170.19, 162.80, 157.40, 155.05, 154.06, 141.82, 140.25, 124.59, 122.45, 115.62 (JC-F = 22.1), 114.40 (JC-F = 22.7), 113.49 (JC-F = 7.3), 104.60 (2C), 100.72, 61.40, 56.81 (2C), 48.93, 48.56, 48.38, 43.76, 31.69, 26.86, 18.71. HR-ESI-MS (m/z): 482.2097 [M + H]^+^, 504.1880 [M + Na]^+^, 520.1732 [M + K]^+^. Anal. Calcd. for C_26_H_28_FN_3_O_5_ (481.2013 g/mol).

5-((6′-bromo-4′-(2′’-oxopyrrolidin-1′’-yl)-3′,4′-dihydroquinolin-1′(2′H)-yl)methyl)-3-phenylisoxazole (6y): 95% yield; Beige solid; IR (ATR): 2949, 2877, 1676, 1497, 1277 cm^−1^. ^1^H NMR (400 MHz, CDCl_3_) δ (ppm): 7.74 (2H, d, *J* = 1.6 Hz), 7.42 (3H, br), 7.17 (1H, d, *J* = 8.7 Hz), 6.97 (1H, s), 6.53 (1H, d, *J* = 8.7 Hz), 6.37 (1H, s), 5.42–5.33 (1H, m), 4.59 (1H, d, *J* = 17.4 Hz), 4.53 (1H, d, *J* = 17.4 Hz), 3.61–3.38 (2H, m), 3.28–3.11 (2H, m), 2.56–2.42 (2H, m), 2.23–2.05 (2H, m), 2.05–1.95 (2H, m). ^13^C NMR (100 MHz, CDCl_3_) δ (ppm): 175.29, 169.05, 162.24, 143.71, 131.18, 129.91 (2C), 128.69 (2C), 128.44, 126.57 (2C), 122.13, 113.35, 109.36, 100.15, 47.88, 47.39, 47.33, 43.33, 31.04, 26.14, 18.08. HR-ESI-MS (m/z): 452.0906 [M + H]^+^, 368.8798 [M-C_4_H_7_NO]^+^. Anal. Calcd. for C_23_H_22_BrN_3_O_2_ (451.0895 g/mol).

3-(4-methoxyphenyl)-5-((6′-bromo-4′-(2′’-oxopyrrolidin-1′’-yl)-3′,4′-dihydroquinolin-1′(2′H)-yl)methyl)isoxazole (**6z**): 67% yield; Beige solid; IR (ATR): 2928, 2039, 1680, 1252, 1023 cm^−1^. ^1^H NMR (400 MHz, CDCl_3_) δ (ppm): 7.68 (2H, d, *J* = 7.4 Hz), 7.17 (1H, d, *J* = 8.8 Hz), 6.97 (1H, s), 6.94 (2H, d, *J* = 7.4 Hz), 6.53 (1H, d, *J* = 8.4 Hz), 6.30 (1H, s), 5.41–5.34 (1H, m), 4.57 (1H, d, *J* = 17.5 Hz), 4.51 (1H, d, *J* = 17.5 Hz), 3.83 (3H, s), 3.60–3.38 (2H, m), 3.28–3.10 (2H, m), 2.57–2.41 (2H, m), 2.22–2.06 (2H, m), 2.05–1.99 (2H, m). ^13^C NMR (100 MHz, CDCl_3_) δ (ppm): 175.33, 168.76, 161.86, 160.87, 143.76, 131.20, 129.91, 128.00, 122.10, 120.95, 114.09, 113.39, 109.34, 99.92, 55.14, 47.88, 47.42, 47.35, 43.37, 31.06, 26.17, 18.10. HR-ESI-MS (m/z): 482.1091 [M + H]^+^, 504.0896 [M + Na]^+^, 520.0751 [M + K]^+^. Anal. Calcd. for C_24_H_24_BrN_3_O_3_ (481.1001 g/mol).

3-(3,4-dimethoxyphenyl)-5-((6′-bromo-4′-(2′’-oxopyrrolidin-1′’-yl)-3′,4′-dihydroquinolin-1′(2′H)-yl)methyl)isoxazole (6aa): 70% yield; Orange oil; IR (ATR): 2936, 2832, 1677, 1497, 1264, 1019 cm^−1^. ^1^H NMR (400 MHz, CDCl_3_) δ (ppm): 7.35 (1H, s), 7.23 (1H, d, *J* = 8.2 Hz), 7.17 (1H, d, *J* = 8.7 Hz), 6.97 (1H, s), 6.88 (1H, d, *J* = 8.2 Hz), 6.53 (1H, d, *J* = 8.7 Hz), 6.31 (1H, s), 5.40–5.35 (1H, m), 4.59 (1H, d, *J* = 17.4 Hz), 4.50 (1H, d, *J* = 17.4 Hz), 3.92 (3H, s), 3.90 (3H, s), 3.60–3.38 (2H, m), 3.28–3.11 (2H, m), 2.55–2.44 (2H, m), 2.22–2.05 (2H, m), 2.04–2.00 (2H, m). ^13^C NMR (100 MHz, CDCl_3_) δ (ppm): 175.38, 168.84, 161.97, 150.50, 149.09, 143.71, 131.19, 129.85, 122.08, 121.15, 119.79, 113.36, 110.81, 109.34, 109.05, 99.94, 55.82, 55.72, 47.91, 47.44, 47.35, 43.34, 31.04, 26.12, 18.08. HR-ESI-MS (m/z): 512.1180 [M + H]^+^, 534.0969 [M + Na]^+^, 550.0728 [M + K]^+^. Anal. Calcd. for C_25_H_26_BrN_3_O_4_ (511.1107 g/mol).

5-((6′-bromo-4′-(2′’-oxopyrrolidin-1′’-yl)-3′,4′-dihydroquinolin-1′(2′H)-yl)methyl)-3-(3,4,5- trimethoxyphenyl)isoxazole (6ab): 43% yield; Orange oil; IR (ATR): 2939, 2830, 1680, 1419, 1125, 1000 cm^−1^. ^1^H NMR (400 MHz, CDCl_3_) δ (ppm): 7.18 (1H, d, *J* = 8.7 Hz), 6.99–6.94 (3H, br.s.), 6.52 (1H, d, *J* = 8.8 Hz), 6.31 (1H, s), 5.42–5.36 (1H, m), 4.62 (1H, d, *J* = 17.5 Hz), 4.51 (1H, d, *J* = 17.5 Hz), 3.90 (6H, s), 3.87 (3H, s), 3.63–3.40 (2H, m), 3.29–3.12 (2H, m), 2.55–2.45 (2H, m), 2.24–2.07 (2H, m), 2.06–2.02 (2H, m). ^13^C NMR (100 MHz, CDCl_3_) δ (ppm): 176.04, 169.78, 162.81, 154.05, 144.31, 140.24, 131.87, 130.45, 124.50, 122.79, 113.97, 110.06, 104.58, 100.70, 77.16, 61.38, 56.79, 48.67, 48.13, 48.07, 43.93, 31.69, 26.74, 18.75. HR-ESI-MS (m/z): 542.1245 [M + H]^+^. Anal. Calcd. for C_26_H_28_BrN_3_O_5_ (541.1212 g/mol).

### 3.2. Inhibitory Activity against Cholinesterases

The AChE and BChE inhibitory activities of the synthesized compounds were performed using the methodology described by Ellman [26]. AChE (from *Electrophorus electricus*), BChE (from bovine serum), 5,5′-dithiobis-(2-nitrobenzoic acid) (DTNB), acetylthiocholine and butyrylthiocholine iodides (AChI/BChI) were purchased from Sigma-Aldrich. The alkaloid galantamine was used as the reference compound, considering its competitive mode of action as described in the literature [27]. Synthesized compounds were tested at five different concentrations to obtain cholinesterase inhibitory activity values between 5 to 80%. This assay was performed in 96-well plates, where 50 μL of sample were dissolved in phosphate buffer (8 mM K_2_HPO_4_, 2.3 mM NaH_2_PO_4_, 150 mM NaCl, and 0.05% Tween 20 at pH 7.6) and a solution of 50 μL of AChE/BuChE (0.25 unit/mL) from *Electroporus electricus* and bovine serum, respectively, in the same buffer, was added. The assay solutions, without substrate, were incubated with the enzyme for 30 min at room temperature. After incubation, the substrate was added. The substrate solution consisted of Na_2_HPO_4_ (40 mM), acetylthiocholine/butyrylthiocholine (0.24 mM) and 5,5′-dithio-bis-(2-nitrobenzoic acid) (0.2 mM, DTNB, Ellman’s reagent). Absorbance of the yellow anion product, due to the spontaneous hydrolysis of substrate, was measured at 405 nm for 5 min on a Microtiter plate reader (Multiskan EX, Thermo, Vanta, Finland). The AChE/BuChE inhibition was determined for each compound. The enzyme activity was calculated as a percentage compared to a control sample using only the buffer and enzyme solution. Each assay was run in triplicate and each reaction was repeated at least three independent times. The IC_50_ values were calculated by means of regression analysis.

### 3.3. Enzymatic Kinetic Study

For enzymatic kinetic studies, the enzyme was pre-incubated with different substrate concentrations ranging from 3.75 × 10^−3^ to 0.48 mM. For the determination of type of inhibition, *V_max_* and *K_m_* (Michaelis constant), double reciprocal plots (1/V versus 1/[S] where V = reaction rate and S = substrate concentration) were constructed, using Lineweaver–Burk methods. Determinations were made in the absence and presence of test compounds. For 5n, the concentrations were set to 0, 2.12, and 4.24 μM and for 6aa 0, 1.985, and 3.97 μM. The enzymatic reaction was extended to 5 min before the determination of the absorption. The *V_max_* and *K_m_* values of the Michaelis–Menten kinetics were calculated by nonlinear regression from substrate–velocity curves. Data analysis was carried out with Sigmaplot v10.0 and Enzyme kinetic v1.3 add-on (Systat Software, Inc, Richmond, CA, USA) [28].

### 3.4. Statistical Analysis

All biological assays were performed in triplicate by independent assays, and obtained values were analyzed and expressed as mean standard error of the mean (SEM) using Statistical Product and Service Solutions, 17th version (SPSS, Inc., Chicago, IL, USA)). IC_50_ values for hybrid compound derivatives against AChE were calculated using regression analysis. Statistical analyses were performed by one-way analysis of variance (ANOVA).

### 3.5. Protein and Ligand Structure Preparation

The tridimensional X-ray structures of AChE were obtained from the Protein Data Bank (PDB). As of May 2017, there were sixty-four structures from *Tetronarce californica* (Pacific electric ray) with a resolution less than or equal to 2.5 Å. PDB entries having non-drug or covalent ligands, non-standard residues, and/or missing loops near the active site were excluded, leading to a total of thirty-one structures. The selected crystal structures were further clustered down while keeping those having unique conformations of the amino acids at the active site, thus obtaining only ten crystals with PDB codes: 1E66, 1EA5, 1GPK, 1ODC, 1ZGC, 2C5G, 2CEK, 2CMF, 2CKM, and 2XI4. These structures were processed with the Protein Preparation Wizard module [29], which encompasses bond order assignment, the addition of hydrogen atoms, and prediction of the protonation states of polar residues. X-ray waters beyond 5.0 Å of the corresponding co-crystallized ligand were deleted. Non-conserved X-ray water molecules were also deleted, where one water was considered as conserved if it was present in more than 80% of the initial sixty-four structures. Residues with alternative positions (mostly in the protein surface) were set to that of highest occupancy. Ionization and tautomeric states of the co-crystallized ligands were predicted using Epik [29,30,31] at pH = 7 and set to those of lowest penalty. Then, the hydrogen-bond network was optimized at neutral pH by sampling Asn and Gln rotamers, hydroxyl and thiol terminal groups, and water orientations. The protonation states for residues Asp, Glu, Lys and His were predicted using PropKa 3.7 [32,33]. The structures were then relaxed by means of a restrained molecular minimization using the impact refinement module [34] with the OPLS3 force field [35], with heavy atoms restrained to the initial coordinates within a root-mean-square deviation (RMSD) of 0.30 Å. The compound structures were sketched using Maestro software [36] and then prepared with the LigPrep module [29], where ionization and tautomeric states were generated at pH = 7.0 ± 2.0 using Epik. All possible enantiomers were included, summing up to 120 structures.

### 3.6. Molecular Cross-Docking

All molecular docking calculations were performed using the Glide program [37] with the standard precision (SP) algorithm. As the chemical structure of the synthesized compounds resemble none of the co-crystallized ligands, we employed the ensemble docking approach to take into account the protein flexibility, where the compounds are docked against multiple rigid protein structures. Using multiple protein conformations often increases the success rate of the pose prediction as some compounds may score poorly against some protein conformations but they bind positively to another protein conformation [38]. Such knowledge is difficult to know *a priori*, therefore careful consideration of the protein structures during docking experiments is required. We followed a similar approach based on a previous work on several PPARγ agonist series [39]. Docking grids for the ten selected AChE crystal structures were generated with the default settings using the corresponding cognate ligand as centroid while ensuring that the grid box size was big enough to cover the entire active site. Default docking parameters were used with the following options enabled: enhance planarity of conjugated π groups, include aromatic carbons as H-bond donors, include halogens as halogen bond donors, and add strain correction in the post-docking score. For a comprehensive pose sampling, one-hundred poses were retained per ligand with the root-mean-square deviation (RMSD) criteria for duplicate pose elimination reduced from 0.5 to 0 Å. It was recently shown that these options can improve the identification of the correct pose by enhancing ligand sampling [40]. The obtained poses were clustered by RMSD with a threshold of 2.0 Å with the protein structures aligned beforehand, where only compounds with the same chirality were grouped together. Then, the five most populated clusters with at least ten poses per ligand were selected for pose rescoring. For each cluster, the representative poses (one per ligand) were selected according to the best fit between the calculated energies and the experimental values. Finally, the cluster with the highest coefficient of determination (R^2^) was chosen out of the top five and further analyzed.

### 3.7. Pose Re-Scoring with MM/GBSA

Free binding energies (ΔGbind) for the poses from selected clusters were calculated using the more computationally intensive MM/GBSA method with the OPLS3 force field and the variable dielectric implicit solvent model VSGB 2.0 [41] as implemented in the Prime program [29]. The MM/GBSA method considers that the energy of a molecular system can be approximated by the sum of the energy from molecular mechanics (MM) in gas phase, polar continuum solvation energy (GB), and nonpolar solvation energy obtained from the solvent accessible surface area (SASA). Several investigations have used MM/GBSA to re-score docking poses in virtual screening experiments with great success [42,43,44,45].

### 3.8. ADME Predictions

The pharmacokinetic properties of hybrid compounds were predicted through ADME descriptors using QikProp [29]. Some of the descriptors predicted were molecular weight, van der Waals, surface areas of polar nitrogen and oxygen atoms, H-bond acceptors, H-bond donors, log *p* (octanol/water) based on the Lipinski´s rule of 5.

## 4. Conclusions

In conclusion, a series of THQ-isoxazole/isoxazoline hybrids with different patterns of substitution were designed based on previous report and obtained using a friendly and economical synthetic chemical protocol under mild conditions using the 1,3-dipolar cycloaddition reaction to obtain inhibitors of cholinergic enzymes. Most of compounds showed significant inhibitory effects against AChE and BChE. In general, the tendency of selectivity of the compounds of both synthesized series was moderate. However, some compounds of the series 5 (5k, 5m, 5n, and 5o) showed significant selectivity for inhibiting AChE, while compounds of the series 6 (6a, 6h, 6l, 6u, 6aa) were more selective against to BChE. Therefore, compound 6aa could be introduced as a candidate for further modification in the development of anti-AD drugs. The mode and free binding energies of the most active compound against AChE were analyzed through dual computational protocol using the molecular docking algorithm from the Glide module and the MM/GBSA method from Prime module. Similarly, the structure–activity relationships of the molecular hybrids were established by comparing their computational ∆Gpred values against the biological activities measured experimentally. A good correlation was obtained between the predicted values of ΔGbind and experimental inhibitory activity of the compounds against the AChE enzyme. The analysis of obtained IC_50_ values, the binding modes and calculated affinities as well as the pharmacokinetic properties, all calculated *in silico* for this series of compounds, let us infer that most active compounds reported in this work can be used as new modulators of cholinesterases. Efforts are ongoing to further optimize and provide a systematic structure–activity analysis at the proper position. The results obtained in this work show that the design of hybrid compounds from propargyl fragment as a versatile entity, allows the development of new synthetic molecules with structural diversity, enhancing the biological activity of propargylated tetrahydroquinolines previously described by our work group [19].

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
