# Peer review of "Tetrahydroquinoline-Isoxazole/Isoxazoline Hybrid Compounds as Potential Cholinesterases Inhibitors: Synthesis, Enzyme Inhibition Assays, and Molecular Modeling Studies"

_ijms, 2019, doi:10.3390/ijms21010005_

Round 1
Reviewer 1 Report
The authors addressed all of my previous concerns.
Reviewer 2 Report
This version has been revised as suggested.
This manuscript is a resubmission of an earlier submission. The following is a list of the peer review reports and author responses from that submission.
Round 1
Reviewer 1 Report
Journal: International Journal of Molecular Sciences
Manuscript ID: ijms-619695
Tetrahydroquinoline-Isoxazole/Isoxazoline Hybrid Compounds as Potential Cholinesterases Inhibitors: Synthesis, Inhibition Bioassays, and Molecular Modeling Studies
This paper describes the chemical synthesis and mainly simulation studies of the new class of cholinesterase inhibitors. Although there were no in vivo studies, for example, inhibition assays using culture cells or mice, the presented data seems to be useful for the development of new cholinesterase inhibitors, which may be effective for the treatment of Alzheimer's disease. Since the experiments were performed carefully, this work is worth being published in the Journal. Some suggestions are described below.
<Major Point>
(1) Galantamine as a control of computational models
To evaluate the results using computational models, the data of already established compounds are very useful. In Table 1 and 2, the data of galantamine (the bottom line) help very much to estimate the value of Comp 5n and 6aa. In other results using computational models (Fig 3, Table 3, and Table 4), the data of galantamine should be added. The accuracy of the calculated oral absorption is especially important for the clinical trials.
(2) Galantamine as a control of enzyme assays.
In the experiments of Fig 2 and Fig 4, including the data of galantamine would more definitely show the value of the new inhibitors.
<Minor Points>
(a) Line 31: correlation between the experimental inhibition values and the predicted deltaGbinding.
"deltaGbinding" should be "binding free energy"
(b1) Line 4: Synthesis, Inhibition Bioassays, and Molecular
"Inhibition Bioassays" should be "Enzyme Inhibition Assays", because no in vivo assay (using culture cells or animals) was performed.
(b2) Line 23: Bioassays studies showed that some hybrids exhibited significant potency to inhibit
"Bioassays" should be "Enzyme inhibition assays", for the same reason as above.
(b3) Line 139: Results of bioassays of inhibitory activity of THQ?Isoxazoline hybrids 5a-p on AChE and BChE 1
"bioassays of inhibitory activity" should be "enzyme inhibition assays", for the same reason as above.
(c) In Table 1, 2, AChE (microM) and BChE (micoroM) in the top line of the table should be IC50 (AChE)(microM) and IC50 (BChE)(microM) respectively.
End of File
Reviewer 2 Report
The manuscript by Rodríguez Núñez et al. aims to synthesize and evaluate tetrahydroquinoline-isoxazole/isoxazoline hybrid compounds in the inhibition of cholinesterases (AChE, BChE) as potential drug candidates for Alzheimer’s disease. The manuscript in my opinion does merit publication after major revision. My comments are as follows:
1. It seems to be an accurate experimental work, however, the found IC50 data do not really exceed the ones reported previously in the literature. Their potential advantages (if any) should be discussed in more details in the Discussions or Conclusions section. In general, the obtained data should be discussed and interpreted better throughout the manuscript to support the conclusions drawn at the end.
2. A short experimental description of the enzyme activity measurements and kinetic studies should be given even if they follow previously published and cited protocols. The source of the enzymes should be clearly specified.
3. “40 mostly new hybrid compounds”. What does it mean? Which ones are the novel compounds?
4. Fig. 4: I would call R2=0.83 “very strong correlation”.
5. Molecular docking should be carried out with BChE as well.
6. Conclusions:
a. “Most of compounds showed positive inhibitory effects against AChE and BChE”. I recommend changing ‘positive’ to ‘significant’.
b. “The THQ-isoxazoline hybrids showed selectivity for AChE. However, THQ-isoxazole hybrids were more selective for BChE, and they might have a therapeutic advantage for the treatment of AD.” These are unclear statements, should be explained more (see comment 1).
7. The list of references should be updated with more current (2019) publications behind the authors’ own.